# SELF-BOOSTING LARGE LANGUAGE MODELS WITH SYNTHETIC PREFERENCE DATA

**Qingxiu Dong**[†]   **Li Dong**[‡]   **Xingxing Zhang**[‡]   **Zhifang Sui**[†]   **Furu Wei**[‡]
[†]State Key Laboratory of Multimedia Information Processing,
School of Computer Science, Peking University   [‡]Microsoft Research

## ABSTRACT

Through alignment with human preferences, Large Language Models (LLMs) have advanced significantly in generating honest, harmless, and helpful responses. However, collecting high-quality preference data is a resource-intensive and creativity-demanding process, especially for the continual improvement of LLMs. We introduce SynPO, a self-boosting paradigm that leverages synthetic preference data for model alignment. SynPO employs an iterative mechanism wherein a self-prompt generator creates diverse prompts, and a response improver refines model responses progressively. This approach trains LLMs to autonomously learn the generative rewards for their own outputs and eliminates the need for large-scale annotation of prompts and human preferences. After four SynPO iterations, Llama3-8B and Mistral-7B show significant enhancements in instruction-following abilities, achieving over 22.1% win rate improvements on AlpacaEval 2.0 and ArenaHard. Simultaneously, SynPO improves the general performance of LLMs on various tasks, validated by a 3.2 to 5.0 average score increase on the well-recognized Open LLM leaderboard.

## 1 INTRODUCTION

Large Language Models (LLMs) have made remarkable progress in following user instructions and generating honest, harmless, and helpful responses (Achiam et al., 2023; Dubey et al., 2024). This advancement is primarily achieved in the model alignment stage, which involves training reward models or LLMs directly on datasets curated from human preferences (Ouyang et al., 2022b; Bai et al., 2022a) , typically employing Reinforcement Learning from Human Feedback (RLHF) (Ouyang et al., 2022b) or Direct Preference Optimization (DPO) (Rafailov et al., 2024).

Recent research has made significant strides in model alignment by collecting high-quality preference data (Hu et al., 2024) , sampling and ranking on-policy responses (Meng et al., 2024; Wu et al., 2024b), or introducing LLM-as-a-Judge as substitutes for human preferences (Yuan et al., 2024; Cui et al., 2023). However, most work still relies on static, pre-collected preference datasets from human or stronger LLM annotation. As LLMs improve rapidly, collecting large, high-quality preference data for effective learning becomes increasingly challenging and costly, whether from humans or stronger models (Shi et al., 2023). According to Yin et al. (2024), directly sampling preference pairs, which closely resembles an on-policy setting, can result in performance declines due to inherent volatility and inefficiency. Therefore, constructing effective preference data to continuously improve LLMs remains a critical research problem.

In this work, we present a self-boosting paradigm for LLM alignment, SynPO. This paradigm leverages a small set of supervised fine-tuning (SFT) data to steer the generation of synthetic preference data, thereby enabling LLMs to iteratively extend their capabilities through optimizing on synthetic data. To support iterative preference learning across diverse scenarios, SynPO first trains a self-prompt generator to create large-scale synthetic prompts. Unlike previous approaches that require more powerful LLMs and instruction examples (Wang et al., 2022), our generator utilizes only the LLM itself and three random keywords as input. To generate preference pairs for the synthetic prompts, SynPO utilizes the initial model generated responses as rejected candidates and employs a response improver to refine these responses into chosen ones. The response improver comes from two straightforward intuitions: (1) LLMs excel at identifying distribution gaps between texts (Zhong

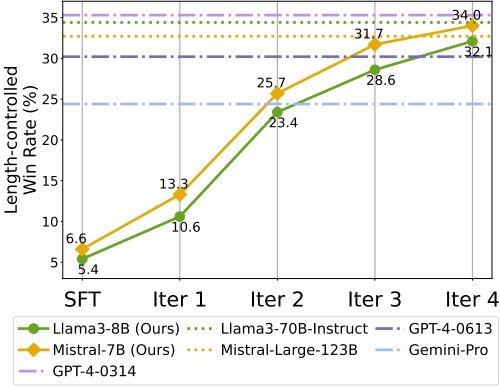

Figure 1: Length-controlled win rate on AlpacaEval 2.0 improves with SynPO iterations, approaching GPT-4 level for the base versions of Llama3-8B and Mistral-7B.

| Model | Size | LC (%) | WR (%) |
|---|---|---|---|
| gpt4_1106_preview | - | 50.0 | 50.0 |
| GPT-4 (03/14) | - | 35.3 | 22.1 |
| Meta-Llama-3-70B-Instruct | 70B | 34.4 | 33.2 |
| **Mistral-Base-SynPO** *Iter4* | **7B** | **34.0** | **36.4** |
| Mistral Large (24/02) | 123B | 32.7 | 21.4 |
| **Mistral-Base-SynPO** *Iter3* | **7B** | **31.7** | **33.8** |
| GPT-4 (06/13) | - | 30.2 | 15.8 |
| Claude 2 | - | 28.2 | 17.2 |
| Claude 2.1 | - | 27.3 | 17.0 |
| **Mistral-Base-SynPO** *Iter2* | **7B** | **25.7** | **28.1** |
| gemini-pro | - | 24.4 | 18.2 |
| Mixtral-8x7B-Instruct-v0.1 | 8x7B | 23.7 | 18.3 |
| Mistral-7B-Instruct-v0.2 | 7B | 17.1 | 14.7 |
| **Mistral-Base-SynPO** *Iter1* | **7B** | **13.3** | **15.3** |
| Mistral-Base-SFT | 7B | 6.6 | 3.6 |

Table 1: Results on AlpacaEval 2.0 leaderboard. LC and WR represent length-controlled and raw win rate, respectively.

et al., 2022; Singh et al., 2022), and (2) refining a response is generally easier than generating a high-quality response from scratch (Madaan et al., 2023; Lu et al., 2023; Nguyen et al., 2024). In each iteration, we train the same initial model to be a response improver, focusing on identifying distribution gaps between current model outputs and gold standard responses in seed data. We then use the response improver to refine the initial model outputs, thereby providing generative rewards to the responses. Through iterative training the initial model on synthetic preference data, SynPO allows the LLM to make subtle improvements and gradually push its boundaries. By leveraging small high-quality data and the current model state to guide the generation of synthetic data, we introduce stronger supervision in an iterative manner.

Experimental results demonstrate that SynPO not only benefits LLM alignment with human preferences, but also improves generalist capabilities across various tasks. Trained solely on synthetic data, SynPO significantly improves the instruction-following abilities of Llama3-8B and Mistral-7B (as shown in Figure 1 and Table 1), achieving over a 26% length-controlled win rate improvement on AlpacaEval 2.0 (Dubois et al., 2024) and a 22% to 30% improvement on Arena-hard (Li et al., 2024c) (as shown in Table 2). Furthermore, self-boosted models achieve 3.2% to 5.0% higher average performance than SFT models on the Open LLM leaderboard (Beeching et al., 2023), indicating SynPO also enhances general LLM performance.

To summarize, our contribution includes:

- We introduce SynPO, a self-boosting mechanism that iteratively induces LLMs to synthesize high-quality data for training. Without requiring human-annotated preference data, SynPO significantly enhances the diversity and quality of synthetic prompts and responses.
- SynPO dynamically guides LLMs to improve their own outputs, using pre- and post-refinement generations as synthetic preference pairs for training. This approach effectively integrates generative rewards for preference learning, enabling LLMs to gradually push their boundaries.
- SynPO significantly enhances both the instruction-following capabilities and the general performance of LLMs, showing substantial improvements over three to four iterations.

## 2 SELF-BOOSTING LLM WITH SYNTHETIC PREFERENCE DATA

SynPO is a self-boosting scheme designed to iteratively generate high-quality preference data. An overview of SynPO is presented in Figure 2. It begins with a small set of SFT data as seed data, denoted as $\{(\mathbf{x}_i^*, \mathbf{y}_i^*)\}_{i=0}^{n}$, and the initial policy model $\pi_{\theta_0}$. By incorporating both the self-prompt generator and the response improver, SynPO provides sufficient prompts for iterative training and leverages the generative rewards in the synthetic preference data. This approach allows the policy model to make subtle improvements and gradually expand its boundaries.

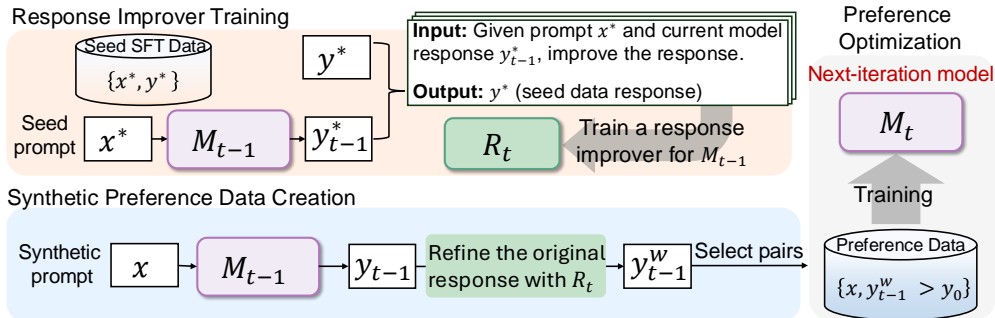

Figure 2: **Overview of SynPO in the $t^{th}$ iteration.** Starting with the previous iteration model $M_{t-1}$, SynPO first learns a response improver $R_t$ to identify discrepancies between model responses $(y_{t-1}^*)$ and gold standard responses $(y^*)$ on seed data, and learns to refine model responses. Subsequently, on the self-generated prompts $x$ (elaborated in Section 2.1), SynPO employs $R_t$ to refine the $M_{t-1}$ responses $(y_{t-1})$ into improved responses $(y_{t-1}^w)$. The valid synthetic prompts $x$, refined responses $(y_{t-1}^w)$, and initial model $M_0$ responses $(y_0)$ to form synthetic preference data. These data are incorporated into the synthetic preference dataset for preference optimization, resulting in an updated $M_t$ for the next iteration. The iterative process continually enhances LLM capabilities in instruction-following and task performance.

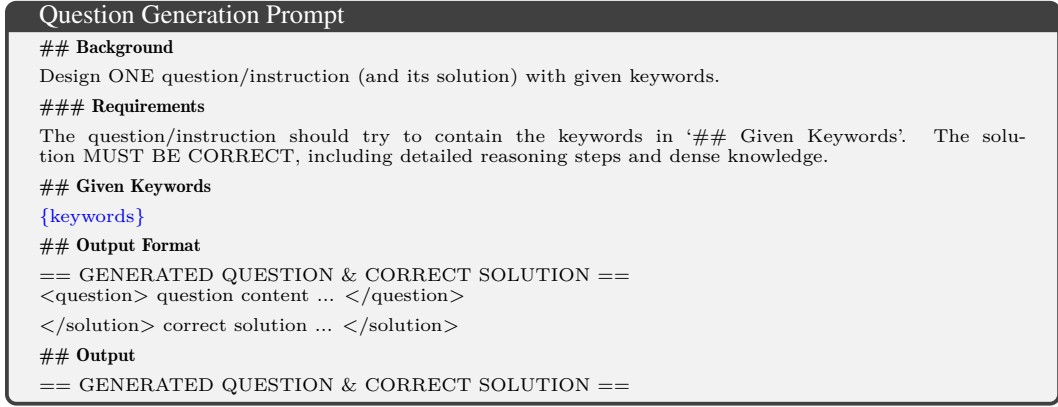

Figure 3: Prompt used in SynPO for LLMs to act as self-prompt generators.

## 2.1 SYNTHETIC PROMPT CREATION

Diverse and ample prompts are crucial for effective preference learning (Shi et al., 2023; Yuan et al., 2024; Song et al., 2024). Diversity facilitates generalization and a sufficient number of prompts allows data selection from a large candidate pool. In SynPO, we propose a novel strategy for synthetic prompt generation. We design a keywords-to-text task to guide the training of a self-prompt generator and create pseudo-label data from the seed SFT data.

**Self-Prompt Generator Training**  We train the LLM itself to serve as a high-quality prompt generator. For each prompt $\mathbf{x}_i^*$ in seed data, we randomly extract two keywords from $\mathbf{x}_i^*$ and one noise keyword from $\mathbf{x}_j^*$, where $j \in \{1, 2, \ldots, n\} \setminus \{i\}$. The inclusion of the noise keyword enhances the robustness of the prompt generator. It learns to filter out irrelevant keywords during training and ensure that the generated prompts are fluent. This process yields a keyword list, $k_i$ for $\mathbf{x}_i^*$. Next, we insert $k_i$ into a prompt template (see Figure 3) to create a prompt and use $(\mathbf{x}_i^*, \mathbf{y}_i^*)$ as the corresponding completion. This process constructs training data for the prompt generator. We then optimize $\theta_0$ through SFT to transform the model into a prompt generator $\mathcal{G}$. $\mathcal{G}$ possesses the capability to generate unlimited, diverse, and high-quality user instructions, controlled by the given keywords.

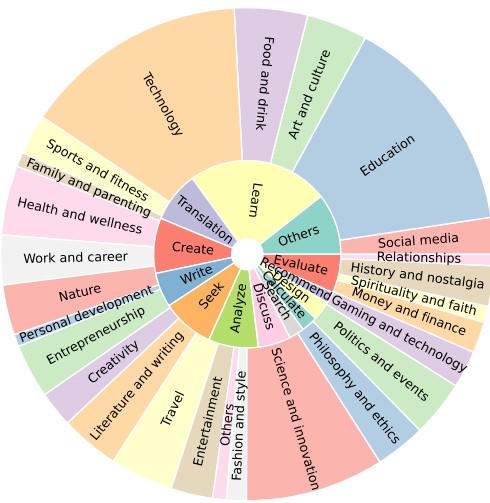

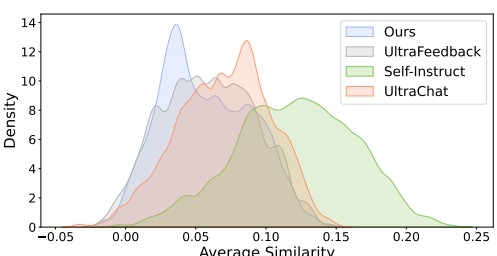

Figure 4: The top 25 most common topics (outer circle) and the top 12 most common intentions (inner circle) in SynPO generated prompts. We aggregate the other topics and intentions to the 'Others' group.

Figure 5: Inter-prompt similarity distributions for 1,000 randomly sampled prompts from SynPO, UltraFeedback (Cui et al., 2023), Self-Instruct (Wang et al., 2022), and Ultra-Chat (Ding et al., 2023). We used Sentence-Transformer (Reimers & Gurevych, 2019) to compute sentence embeddings and calculated the cosine similarity between each prompt and all others, then averaged these values for each prompt. The results suggest that our method, SynPO, generates more diverse prompts with lower inter-prompt similarity.

**Keywords Sampling and Prompt Generating** To enhance the overall diversity of prompts, we sample keywords from pretrain corpus paragraphs. We select three keywords from a same paragraph to maintain the inherent distribution between keywords, and sample from different paragraphs to ensure overall diversity. Specifically, we randomly sample keyword lists from the RefinedWeb paragraphs (Penedo et al., 2023) and generate $m$ synthetic prompts, denoted $\{\mathbf{x}_i\}_{i=1}^m$, $\mathbf{x}_i \sim \mathcal{G}(\cdot|\mathbf{k}_i)$.

Benefiting from the self-prompt generator, our approach requires only a small SFT data to enable the model to generate diverse prompts independently, no in-context learning examples or predefined topic lists are required. Due to the high combinatorial possibilities of keyword sampling, the self-prompt generator can produce a huge variety of synthetic prompts for preference learning.

For further analysis, we randomly sampled 1k self-generated prompts from the Llama3-8B-Base prompt generator and used GPT-4 Turbo to classify the intentions and topics behind these prompts[1]. The results, shown in Figure 4, demonstrate significant diversity across various topics and user intentions. Even compared to prompts from GPT3.5-Turbo (Ding et al., 2023) or a collection of prompts from different sources (Cui et al., 2023), as shown in Figure 5, SynPO generated prompts exhibit lower inter-prompt similarity and greater diversity.

## 2.2 SYNTHETIC PREFERENCE GENERATION

A primary challenge in leveraging synthetic prompts is the lack of high-quality responses to provide sufficient supervision (Li et al., 2024b). To address this, we introduce a response improver to enhance the quality of model responses to synthetic prompts. Pre- and post-improvement responses naturally become the rejected and chosen candidates, respectively, with the chosen ones providing clear guidance on what approximates a gold standard response.

**Response Improver Training** In each iteration, we train the LLM as a response improver to further reduce the gap between policy model outputs and gold standard responses. Formally, let $\pi_{\boldsymbol{\theta}_{t-1}}$ denote the policy model at the beginning of the $t$-th iteration. We generate outputs from $\pi_{\boldsymbol{\theta}_{t-1}}$ for the seed data prompts: $\mathbf{y}^*_{(t-1),i} \sim \pi_{\boldsymbol{\theta}_{t-1}}(\cdot|\mathbf{x}^*_i)$, $i \in \{1, \ldots, m\}$. These outputs, along with the seed data responses, form the training set for the response improver, following the template provided in Appendix B. Each training example consists of the prompt and the policy model output $(\mathbf{x}^*_i, \mathbf{y}^*_{(t-1),i})$ as the input, and the gold standard response $\mathbf{y}^*_i$ as the output. We fine-tune $\pi_{\boldsymbol{\theta}_0}$ on the

---

[1]Experimental details in Appendix F.

training set to obtain the response improver $\mathcal{R}_t$. This response improver refines the policy model outputs, aligning them more closely with the gold standard responses.

**Response Improving** Subsequently, we use $\mathcal{R}_t$ to refine model responses to synthetic prompts, obtaining pre- and post-improvement responses as synthetic preference pairs. For each synthetic prompt $\mathbf{x}_i$, we first obtain the current model output $\mathbf{y}_{(t-1),i} \sim \pi_{\boldsymbol{\theta}_{t-1}}(\cdot|\mathbf{x}_i)$, for $i \in \{1, \ldots, m\}$. The response improver then refines this completion to produce $\overline{\mathbf{y}_{(t-1),i}} \sim \mathcal{R}_t(\cdot|\mathbf{x}_i, \mathbf{y}_{(t-1),i})$, considered the chosen response. As we fine-tune the initial model in each iteration, the initial policy model output $\mathbf{y}_{(0),i}$ serves as the on-policy rejected response for $\mathbf{x}_i$. Here, $\mathbf{y}_{(0),i} \sim \pi_{\boldsymbol{\theta}_0}(\cdot|\mathbf{x}_i)$. This method generates numerous synthetic preference candidates, including both chosen and rejected responses.

**Data Filtering** Unlike data from humans or strong teacher LLMs, which come with clear standard responses, self-generated data require proper filtering to ensure quality (Gulcehre et al., 2023). As our policy model improves, many responses no longer need refining; we only need to retain data with a preference gap between the chosen and rejected responses. Instead of using GPT4-Turbo-as-a-Judge for data filtering (Rosset et al., 2024), SynPO employs only a small model (e.g., a 0.4B PairRM (Jiang et al., 2023)) or the model itself for scoring. Similar to SimPO (Meng et al., 2024) and SPPO (Wu et al., 2024b), this ensures the process does not rely on a more powerful teacher model. We retain $\overline{\mathbf{y}_{(t-1),i}}$ and $\mathbf{y}_{(0),i}$ with significant preference differences (i.e., a large score gap). $\overline{\mathbf{y}_{(t-1),i}}$ is regarded as the chosen response, $\mathbf{y}_i^w$, while $\mathbf{y}_{(0),i}$ is regarded as the rejected response, $\mathbf{y}_i^l$. Along with the corresponding prompt $\mathbf{x}_i$, they form a valid instance $(\mathbf{x}_i, \mathbf{y}_i^w, \mathbf{y}_i^l)$. All valid data is then integrated into the synthetic preference data for subsequent iterations.

## 2.3 SYNTHETIC PREFERENCE OPTIMIZATION

The large-scale synthetic preference data naturally facilitate the multi-iteration process of self-boosting. In each iteration, we follow SimPO (Meng et al., 2024) for training; actually, our method is also compatible with other preference optimization training methods, such as DPO (Rafailov et al., 2024) and KTO (Ethayarajh et al., 2024). Denoting $\mathcal{D}$ as the synthetic preference data, we have:

$$\boldsymbol{\theta}_t \leftarrow \arg\min_{\boldsymbol{\theta}} \mathbb{E}_{(\mathbf{x}_i,\mathbf{y}_i^w,\mathbf{y}_i^l)\sim\mathcal{D}} \left[ \log \sigma \left( \frac{\beta}{|\mathbf{y}_i^w|} \log \pi_{\theta_{t-1}}(\mathbf{y}_i^w \mid \mathbf{x}_i) - \frac{\beta}{|\mathbf{y}_i^l|} \log \pi_{\theta_{t-1}}(\mathbf{y}_i^l \mid \mathbf{x}_i) - \gamma \right) \right]$$

$\beta$ and $\gamma$ are hyperparameters. Different from the vanilla SimPO, SynPO is a iterative process and all the preference data are synthetic ones. The response improver continuously refines the generation distribution to align with the ideal data distribution across multiple iterations.

Overall, the response improver automatically learns to generate implicit generative rewards for the outputs of the LLM. Unlike using a discriminative reward model straightforwardly, this approach helps the model learning to improve its outputs. We present the *Synthetic Preference Optimization* algorithm in Appendix A. The entire optimization process is performed on synthetic preference data, requiring only a small amount of high-quality data for validation. This strategy maintains two key advantages: (1) Compared to the limited and hard-to-collect preference data, SynPO generates an unlimited amount of new self-synthetic data to meet the needs of iterative model improvement. (2) Using small, high-quality validation data prevents the model from deviating during training and consistently guides the generation of more relevant synthetic data.

## 3 EXPERIMENTS

We carry out comprehensive experiments to demonstrate the effectiveness of SynPO in enhancing model alignment and improving general model performance.

### 3.1 EXPERIMENTAL SETUP

**Models and Training** We perform synthetic preference optimization on both Mistral-Base 7B and Llama3-8B Base. Following Meng et al. (2024), we employ supervised fine-tuned models as the initial models. Specifically, the Mistral-Base 7B model (mistralai/Mistral-7B-v0.1) and the Llama3-8B Base model (meta-llama/Meta-Llama-3-8B-Base) were fine-tuned on the UltraChat-200k dataset

| Data Construction | Mistral-Base (7B) | | | Llama3-Base (8B) | | |
|---|---|---|---|---|---|---|
| | AlpacaEval 2.0 | | Arena-Hard | AlpacaEval 2.0 | | Arena-Hard |
| | LC (%) | WR (%) | WR (%) | LC (%) | WR (%) | WR (%) |
| SFT | 6.6 | 3.6 | 2.0 | 5.4 | 3.1 | 2.7 |
| Manual Collection | 21.5 | 20.8 | 16.8 | 22.0 | 19.8 | 23.2 |
| Sampling-Ranking  *Iter1* | 6.5 | 4.4 | 4.1 | 7.2 | 4.3 | 4.4 |
| Sampling-Ranking  *Iter2* | 9.3 | 6.4 | 4.3 | 7.7 | 4.7 | 6.2 |
| Sampling-Ranking  *Iter3* | 10.6 | 7.5 | 7.9 | 13.8 | 8.2 | 8.4 |
| Sampling-Ranking  *Iter4* | 11.6 | 8.0 | 9.6 | 14.2 | 8.4 | 10.4 |
| Self-Rewarding  *Iter1* | 19.5 | 19.8 | 11.9 | 20.1 | 20.3 | 20.8 |
| Self-Rewarding  *Iter2* | 22.4 | 23.5 | 19.2 | 21.7 | 22.4 | 20.5 |
| Self-Rewarding  *Iter3* | 24.6 | 26.3 | 20.8 | 22.4 | 24.1 | 23.8 |
| Self-Rewarding  *Iter4* | 26.1 | 28.0 | 21.1 | 24.8 | 25.6 | 25.0 |
| SYNPO  *Iter1* | 13.3 | 15.3 | 9.8 | 10.6 | 10.7 | 11.6 |
| SYNPO  *Iter2* | 25.7 | 28.1 | 20.8 | 23.4 | 24.1 | 24.6 |
| SYNPO  *Iter3* | 31.7 | 33.8 | **24.1** | 28.6 | 31.5 | **32.5** |
| SYNPO  *Iter4* | **34.0** | **36.4** | 22.8 | **32.1** | **33.6** | 31.4 |

Table 2: Results on AlpacaEval 2.0 and Arena-Hard. LC and WR denote length-controlled and raw win rates, respectively. After four SYNPO iterations, Mistral-Base and Llama3-Base increase LC by 27.4% and 26.7%, respectively, on AlpacaEval 2.0. In Arena-Hard, SYNPO achieves the highest WR by the third iteration, improving both models by over 22.1%.

as part of the Zephyr (Tunstall et al., 2023) training pipeline.[2] Subsequently, we utilize 18k seed data to SFT the self-prompt generator and then generate 50k synthetic prompts per iteration. For data filtering, we employ the 0.4B PairRM (Jiang et al., 2023) as a small pairwise scoring model for the Mistral-Base 7B. For Llama3-Base 8B, we use Llama3 itself (ArmoRM-Llama3-8B-v0.1[3]) as a scoring model for data filtering, given its superior alignment with human scoring (Meng et al., 2024). In each iteration $t = 1, \ldots, T$, we use model $\pi_{\theta_{t-1}}$ from the previous iteration to generate synthetic preference data and then preference optimize the initial models again. More details on training parameters, filtering thresholds, and implementation environments are provided in Appendix C.

**Seed Data Construction**   We randomly sample UltraFeedback (Cui et al., 2023) prompts and their GPT-4 Turbo completions as our seed data. The seed data is multipurposely transformed for the training of self-prompt generator, response improver, and the validation of synthetic preference optimization. The complete UltraFeedback dataset contains 61k instructions from sources including TruthfulQA (Lin et al., 2021), FalseQA (Hu et al., 2023), Evol-Instruct (Xu et al., 2023a), Ultra-Chat (Ding et al., 2023), and ShareGPT (Chiang et al., 2023). To construct seed SFT data with high-quality responses, we randomly sampled 18k prompts and obtained the corresponding completions generated by GPT-4 Turbo.

**Baselines**   As baselines, we use the initial supervised fine-tuned models and those optimized with data from various preference construction methods, including manual collection and iterative approaches. We recognize the UltraFeedback preference data (Cui et al., 2023) as a product of manual collection. It is gathered from six high-quality datasets and various models, with preferences annotated by GPT-4 (Achiam et al., 2023). For iterative construction approaches, we compare against Sampling-Ranking and Self-Rewarding. For Sampling-Ranking, similar to Meng et al. (2024) and Wu et al. (2024b), we use LLMs in sampling five responses per prompt in each iteration. The same scoring models, i.e., PairRM and ArmoRM-Llama3-8B-v0.1, are then used to select the highest and lowest scoring responses as the chosen and rejected responses, respectively. For the Self-Rewarding method (Yuan et al., 2024), we generate preference data based on model's own rewards via LLM-as-a-Judge prompting. We employ our 18k seed data as the initial instruction-following data. Given that Self-Rewarding requires additional LLM-as-a-Judge training data, we generate 16k

---

[2] https://huggingface.co/alignment-handbook/zephyr-7b-sft-full and https://huggingface.co/princeton-nlp/Llama-3-Base-8B-SFT

[3] https://huggingface.co/RLHFlow/ArmoRM-Llama3-8B-v0.1

| Data Construction | Mistral-Base | | Llama3-Base | |
|---|---|---|---|---|
| | Turn 1 | Turn 2 | Turn 1 | Turn 2 |
| SFT | 6.04 | 5.65 | 6.55 | 5.36 |
| Manual Collection | 6.73 | **6.82** | 7.29 | 7.00 |
| Sampling-Ranking *Iters** | 6.83 | 6.18 | 7.06 | 6.99 |
| Self-Rewarding *Iters** | 6.71 | 6.63 | 7.30 | 7.28 |
| SYNPO *Iter1* | 6.53 | 6.41 | 6.99 | 6.65 |
| SYNPO *Iter2* | 6.66 | 6.65 | 7.34 | 7.30 |
| SYNPO *Iter3* | **6.86** | **6.82** | 7.34 | **7.34** |
| SYNPO *Iter4* | 6.73 | 6.69 | **7.43** | 7.04 |

Table 3: Multi-turn evaluation on MT-Bench. An asterisk (*) denotes the best score across multiple iterations. For Sampling-Ranking, Llama's best is from iteration 4 and Mistral's from iteration 3. For Self-Rewarding, both are from iteration 3. SYNPO progressively enhances the multi-turn instruction-following capabilities of LLMs.

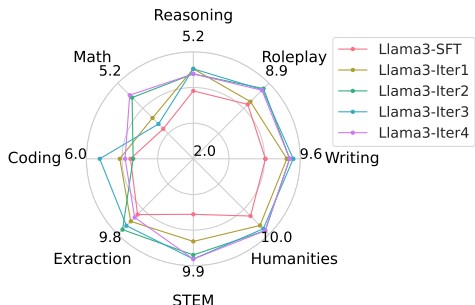

Figure 6: Radar chart for Llama3-8B-Base-SYNPO on MT-Bench. SYNPO achieves notable improvements across various prompt categories, particularly in RolePlay, STEM, Reasoning, and Coding tasks.

seed data with GPT-4 Turbo. For all settings, we adopt SimPO (Meng et al., 2024) for preference optimization. More detailed are elaborated in Appendix C.5.

## 3.2 PREFERENCE ALIGNMENT

We evaluate the model alignment performance on three benchmarks: AlpacaEval 2.0 (Dubois et al., 2024), Arena-Hard (Li et al., 2024c), and MT-Bench (Zheng et al., 2024). AlpacaEval 2.0 includes 805 user prompts and utilizes pair-wise comparison with LLM-as-a-Judge. Specifically, the win rate against the baseline GPT-4 Turbo model is determined based on GPT-4 Turbo evaluation. Arena-Hard includes 500 more challenging user queries, employing GPT-4-Turbo to judge the model responses against GPT-4. MT-Bench features 80 multi-turn questions spanning various domains, with GPT-4 scoring the model responses out of 10.[4]

**Single-Turn Dialogues** We compare the instruction-following and human preference alignment capabilities on AlpacaEval 2.0 (Dubois et al., 2024) and Arena-Hard (Li et al., 2024c) in Table 2. Compared to the initial model post-SFT, SynPO shows sustained improvement over four iterations in win rate against GPT-4 Turbo or GPT-4. On AlpacaEval 2.0, Mistral-Base achieves a 27.4% increase in length-controlled win rate and a 32.8% increase in raw win rate after four iterations. Similarly, Llama3 exhibits a 26.7% rise in length-controlled win rate and a 30.5% improvement in raw win rate after the same number of iterations. In the more challenging Arena-Hard setting, SynPO reaches the highest win rate after the third iteration. Compared to the baseline methods, SynPO's iterative preference learning on synthetic data yielded more significant improvements.

**Multi-Turn Dialogues** For the multi-turn benchmark MT-Bench, we report both the first-turn and second-turn scores (in Table 3) as well as a radar chart depicting performance across different question types (refer to Figure 6).[5] The results indicate that SynPO enhances not only first-turn performance, with an increase of over 0.7 points, but also subsequent turns, with an increase of over 1.2 points. Compared to the initial model, SynPO shows improved performance across various question types, particularly in humanities, writing, STEM, and roleplaying.

## 3.3 DOWNSTREAM TASK PERFORMANCE

In terms of the general model performance on various tasks, we report the average scores on the well-recognized Open LLM Leaderboard (Beeching et al., 2023) and 6 additional benchmarks from Language Model Evaluation Harness library (LLM Harness) (Gao et al., 2024). Open LLM Leaderboard (Beeching et al., 2023) is recognized as a standard assessment for the general per-

---

[4]More details on API usage for LLM judgement are listed in Appendix E and G.
[5]Similar results for Mistral-7B are shown in Figure 10 in Appendix G.

| Model | Arc | HellaSwag | TQA | MMLU | Winogrande | GSM8k | Average |
|---|---|---|---|---|---|---|---|
| Mistral-Base-SFT | 58.02 | 80.89 | 40.37 | 58.78 | 76.40 | 34.72 | 58.20 |
| Manual Collection | 62.71 | 83.39 | 50.69 | 58.47 | 77.35 | 32.83 | 60.91 |
| Sampling-Ranking *Iters** | 60.32 | 81.80 | 44.43 | 59.09 | 76.95 | 36.85 | 59.91 |
| Self-Rewarding *Iters** | 60.15 | 81.84 | 43.25 | 58.98 | 76.48 | 34.72 | 59.24 |
| Mistral-Base-SYNPO *Iter1* | 60.49 | 82.25 | 50.36 | 59.00 | 76.48 | 36.39 | 60.83 |
| Mistral-Base-SYNPO *Iter2* | 63.65 | 83.24 | 58.04 | 58.74 | 76.48 | 27.35 | 61.25 |
| Mistral-Base-SYNPO *Iter3* | 63.54 | 83.14 | 58.11 | 58.37 | 75.77 | 25.26 | 60.70 |
| Mistral-Base-SYNPO *Iter4* | 63.57 | 83.04 | 56.12 | 58.75 | 75.77 | 31.08 | **61.39** |
| LLama3-Base-SFT | 60.92 | 81.28 | 45.37 | 63.80 | 76.72 | 51.93 | 63.34 |
| Manual Collection | 66.72 | 82.89 | 59.47 | 63.10 | 77.82 | 45.72 | 65.95 |
| Sampling-Ranking *Iters** | 66.38 | 82.71 | 59.84 | 63.37 | 77.27 | 54.40 | 67.33 |
| Self-Rewarding *Iters** | 64.76 | 82.48 | 55.54 | 63.42 | 77.03 | 54.59 | 66.30 |
| LLama3-Base-SYNPO *Iter1* | 63.99 | 82.66 | 54.20 | 64.02 | 77.51 | 56.10 | 66.41 |
| LLama3-Base-SYNPO *Iter2* | 65.70 | 83.22 | 61.73 | 64.03 | 76.56 | 56.25 | 67.92 |
| LLama3-Base-SYNPO *Iter3* | 66.55 | 83.57 | 63.53 | 63.91 | 76.80 | 55.27 | 68.27 |
| LLama3-Base-SYNPO *Iter4* | 66.47 | 83.44 | 63.69 | 63.79 | 76.90 | 55.72 | **68.34** |

Table 4: Open LLM Leaderboard results. TQA stands for TruthfulQA. The asterisk (*) represents the best performance across multiple iterations. Compared to the SFT versions, SYNPO achieves an overall improvement of 3.19% for Mistral and 5.00% for Llama3 on the average score.

formance of LLMs. It includes six different datasets, evaluating LLMs on commonsense reasoning (Arc (Clark et al., 2018), HellaSwag (Zellers et al., 2019), Winogrande (Sakaguchi et al., 2021)), wide knowledge (MMLU (Hendrycks et al., 2020), TruthfulQA (Lin et al., 2021)), and math (GSM8k (Cobbe et al., 2021)). The six addtional LLM Harness tasks include Openbook Question Answering (OBQA) (Mihaylov et al., 2018) and Haerae (Son et al., 2023) for model knowledge, MathQA (Amini et al., 2019), XNLI (Conneau et al., 2018), and PROST (Aroca-Ouellette et al., 2021) for reasoning, as well as Toxigen (Hartvigsen et al., 2022) for toxicity evaluation.[6]

**Open LLM Leaderboard** On the Open LLM leaderboard, we observe an overall improvement of 3.19% of Mistral-Base-SFT and 5.00% of Llama3-Base-SFT on the average score. Specifically, SynPO achieves over 6% improvement on the ARC challenge and over 16% on TruthfulQA compared to the SFT model, after four rounds of self-boosting. Notably, the performance of Mistral-Base on GSM8K has experienced a decline, whereas Llama3-Base has demonstrated an improvement of nearly 4 points on the same benchmark. This disparity likely stems from the superior data filtering capability of ArmoRM-Llama3-8B-v0.1 compared to the 0.4B PairRM. ArmoRM-Llama3-8B-v0.1 effectively mitigates erroneous responses and enhances mathematical problem-solving performance.

**LLM Harness Tasks** The advantages above are also reflected in the results for more diverse tasks in LLM Harness, as evidenced by Table 5. Previous works (Wu et al., 2024b; Meng et al., 2024) demonstrate that preference optimization can induce the "alignment tax" - aligning models with human preferences can improve performance for only 1∼2 iterations or even degrade overall performance on downstream tasks (Askell et al., 2021). Our method exhibits similar behavior on MathQA; however, overall, SynPO shows improvements across more iterations on other tasks. This is because synthesizing better chosen candidates introduces additional supervision, partially mitigating the alignment tax issue and enabling LLMs to continuously enhance their capabilities on downstream tasks at the same time of alignment.

## 4 ABLATION STUDIES

### 4.1 SYNTHETIC PROMPTS AND RESPONSES

We have demonstrated the diversity of prompts generated by SynPO in Section 2.1. To further validate the self-prompt generator, we compare the generated prompts with manual collected

---

[6]We follow the standard few-shot setting on Open LLM Leaderboard, as elaborated in Appendix G. For the six additional tasks, we employ a fixed 5-shot setting for evaluation.

| Model | | OBQA | Haerae | MathQA | XNLI | Toxigen | PROST | Average |
|---|---|---|---|---|---|---|---|---|
| Mistral-Base-SFT | | 46.40 | 39.96 | 36.25 | 43.76 | 60.11 | 52.04 | 46.42 |
| Manual Collection | | 50.20 | 40.35 | 36.72 | 44.65 | 62.83 | 54.66 | 48.24 |
| Sampling-Ranking | *Iters** | 47.60 | 40.15 | 36.21 | 44.71 | 62.45 | 53.74 | 47.48 |
| Self-Rewarding | *Iters** | 48.60 | 40.15 | 36.25 | 44.17 | 61.28 | 53.29 | 47.29 |
| Mistral-Base-SynPO | *Iter1* | 48.00 | 39.78 | 36.05 | 44.23 | 60.74 | 52.39 | 46.87 |
| Mistral-Base-SynPO | *Iter2* | 50.40 | 40.05 | 36.85 | 43.97 | 60.96 | 53.04 | 47.55 |
| Mistral-Base-SynPO | *Iter3* | 51.20 | 40.51 | 35.88 | 44.37 | 60.74 | 53.45 | 47.69 |
| Mistral-Base-SynPO | *Iter4* | 51.40 | 40.88 | 36.48 | 44.47 | 63.40 | 55.05 | **48.61** |
| Llama3-Base-SFT | | 46.20 | 61.78 | 42.04 | 45.47 | 68.83 | 52.40 | 52.79 |
| Manual Collection | | 51.60 | 61.59 | 42.51 | 44.77 | 74.38 | 55.96 | 55.14 |
| Sampling-Ranking | *Iters** | 50.80 | 62.05 | 42.81 | 46.67 | 74.68 | 55.41 | 55.40 |
| Self-Rewarding | *Iters** | 48.60 | 62.42 | 42.78 | 46.31 | 74.15 | 54.23 | 54.75 |
| Llama3-Base-SynPO | *Iter1* | 48.20 | 61.96 | 42.71 | 46.15 | 71.17 | 53.77 | 53.99 |
| Llama3-Base-SynPO | *Iter2* | 50.80 | 62.60 | 42.75 | 46.34 | 74.04 | 54.85 | 55.23 |
| Llama3-Base-SynPO | *Iter3* | 51.00 | 62.51 | 42.78 | 46.37 | 75.11 | 55.29 | 55.51 |
| Llama3-Base-SynPO | *Iter4* | 52.00 | 62.51 | 42.65 | 46.27 | 75.18 | 56.14 | **55.79** |

Table 5: Downstream performance in each SynPO iteration on six tasks in LM Evaluation Harness. An asterisk (*) represents the best performance across multiple iterations.

prompts (Cui et al., 2023) and Self-Instruct prompts (Wang et al., 2022). For each prompt construction approach, we randomly sample 20k prompts and construct response pairs through both SynPO and Sampling-Ranking. We compare the single iteration results of Llama3-8B. As shown in Table 6, whether through self-refinement or Sampling-Ranking, synthetic prompts generated by SynPO lead to better-aligned models, validating the quality of these prompts. It is worth mentioning that SynPO prompts are even more effective than the superset of its seed training data, UltraFeedback prompts. This increased effectiveness may be attributed to the greater diversity of SynPO prompts, achieved through the keyword sampling process in prompt synthesis. Results of mixing SynPO and manual collected prompts further indicate the potential of SynPO in augmenting existing prompts.

## 4.2 IMPACT OF SEED DATA

SynPO involves training LLMs solely on synthetic preference data while using seed SFT data for validation. To investigate the maximum impact of the seed SFT data, we compare SynPO with the following settings: 1) *Seed SFT:* Directly fine-tuning the LLM using seed data. 2) *Seed PO:* For each prompt in the seed SFT data, using the gold standard response in the seed data as the chosen response and the initial policy model response as the rejected response for preference optimization. 3) *Seed SFT + PO:* To avoid distribution shifts in directly using seed SFT data, we first obtain a model fine-tuned on seed data as in 1), then construct preference data using the model output and gold standard responses. 4) *Seed SFT + PO$^{me}$:* Training on data from 3) for multiple epochs.[7] The results on AlpacaEval 2.0 are presented in Table 7. Among the evaluated methods except for SynPO, setting 3) is most analogous to SynPO, and proves to be the most effective. However, due to the limited quantity of seed data, the improvement is less than that achieved by iterative SynPO on synthetic data. Training under such conditions for multiple epochs does not yield further improvements and even degrades performance. These findings validate that SynPO is a promising approach to construct preference data and maximize the utilization of minimal high-quality data.

## 5 RELATED WORK

**Preference Data Construction** Preference data are triplets consisting of user prompts, user-preferred responses, and non-preferred responses. Acquiring preference data from humans can be resource intensive, often constrained by the data collection platform (Ouyang et al., 2022a) or the cost of human annotation (Bai et al., 2022a; Ethayarajh et al., 2022; Nakano et al., 2021). To alleviate this problem, researchers have started using teacher LLMs, such as GPT-4 (Achiam et al., 2023),

---

[7]We compare the results of 2 epochs and 3 epochs and select the better-performing 2 epochs. More experimental details in Appendix D

| Prompts | SynPO | | Sampling-Ranking | |
|---|---|---|---|---|
| | LC (%) | WR (%) | LC (%) | WR (%) |
| Manual Collection | 23.8 | 25.6 | 7.8 | 6.2 |
| Self-Instruct | 21.7 | 21.4 | 9.1 | 4.6 |
| SynPO | **24.3** | 24.5 | 9.2 | 5.3 |
| SynPO Mix. | 23.4 | **29.4** | **14.2** | **6.9** |

Table 6: Comparison of various prompt generation methods on AlpacaEval 2.0. SynPO Mix. combines SynPO prompts with manually collected prompts.

| Method | LC (%) | WR (%) |
|---|---|---|
| Seed SFT | 20.1 | 19.7 |
| Seed PO | 11.6 | 10.7 |
| Seed SFT + PO | 24.6 | 20.9 |
| Seed SFT + PO[me] | 22.4 | 15.0 |
| SynPO  *Iter4* | **32.1** | **33.6** |

Table 7: Impact analysis of seed data on AlpacaEval 2.0. PO[me] refers to preference optimization over multiple epochs.

to simulate human preferences (Cui et al., 2023; Ding et al., 2023; Huang et al., 2024). Given user prompts and candidate responses, this line of work employs a stronger model to annotate preferences, thereby overcoming the scarcity and constraints of existing preference data (Cui et al., 2023). However, the prompts still need to be collected (Cui et al., 2023), which limits the domain, diversity, and quantity of the data. Both human annotation and the utilization of large teacher model APIs incur substantial costs (Shi et al., 2023). Moreover, relying solely on reward scores or win-lose annotation fails to fully capture the subtleties and complexities of human preferences.

**LLM Self-Boosting**  Previous work has advanced the self-boosting of LLMs by searching for high-reward behaviors (Tian et al., 2024; Zhang et al., 2024), using LLMs as judges to select responses (Yuan et al., 2024; Wang et al., 2024a; Wu et al., 2024a; Kim et al., 2024), and leveraging self-play strategies (Chen et al., 2024; Cheng et al., 2024; Luo et al., 2024; Wu et al., 2024b). These works typically use a fixed set of existing prompts, limiting the LLM ability to learn across wide scenarios. Furthermore, the deterministic reward signals in these methods do not help the model recognize subtle discrepancies between its responses and ideal responses. Prior to our work, Constitutional AI (Bai et al., 2022b) and SELF (Lu et al., 2023) used AI to generate non-deterministic feedback for training. Constitutional AI used refinement data for reward models, while SELF employed GPT-4 for data generation and taught models self-refinement. However, these methods did not utilize comparative information between pre- and post-revision texts for training.

**Synthetic Data for LLMs**  Acquiring human-generated data is costly and time-consuming, leading to the use of synthetic data for LLM training (Wang et al., 2022; Xu et al., 2023a; Li et al., 2024b). Unnatural Instructions (Honovich et al., 2022) and Self-Instruct (Wang et al., 2022) use seed instructions to generate new prompts, while WizardLM (Xu et al., 2023a) and WizardMath (Luo et al., 2023) rewrite these instructions into more complex forms using ChatGPT. Seed topics also produce textbook-like data (Li et al., 2023; 2024b) or self-chat dialogues (Xu et al., 2023b; Ding et al., 2023) for instruction tuning. These methods often require strong LLMs or examples, benefiting from model distillation (Xu et al., 2023a; Li et al., 2024b;a). Our approach uses the model itself to generate prompts and responses without needing carefully designed topics, extending beyond the model's inherent sampling space. This brings generative rewards to the LLM self-boosting process, particularly benefiting initially weaker models. It resembles direct preference knowledge distillation (Li et al., 2024d) but does not rely on large-scale teacher model responses for supervision. Additionally, our work shares a similar spirit with CommonGen (Lin et al., 2020) in generating sentences from keywords and CHEF (Seo et al., 2023) in creating synthetic data for contrastive learning.

## 6  CONCLUSION

We introduce self-boosting LLM with synthetic preference data, SynPO, a method for LLM alignment through iterative training on synthetic data. In SynPO, we innovatively base the entire training process on synthetic data and only employ limited SFT data for validation. SynPO diversifies the prompts and dynamically guides LLMs to improve their own output, using pre- and post-refinement generations as synthetic preference pairs for training in the next iteration. Experimental results show that SynPO leads to significant improvements on both instruction-following capabilities and task performance. This strategy sheds light on high-quality synthetic data generation and self-alignment with minimal supervision, both of which are critical for the continuous development of LLMs.

## 7 ACKNOWLEDGEMENT

This paper is supported by NSFC project 62476009.

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

## LIMITATIONS

Our approach begins with a small SFT dataset and does not necessitate specifically labeled data for training a response improver. We assume that model-generated outputs closely resemble the gold standard responses in the SFT, enabling them to serve as training data for the response improver. This necessitates filtering out pairs where the gold standard is inferior to the model-generated output. Such data can cause the response improver to rewrite text through paraphrasing or substantial alteration, as the training data comprises pseudo pairs rather than minimally edited original responses. While prompting a more powerful LLM to generate rewriting-specific data, as suggested by Lu et al. (2023), can alleviate this, it sacrifices the benefit of learning the distribution gap.

SynPO leverages a small, high-quality dataset repeatedly to guide synthetic data generation, making the seed data quality vital. This approach requires only a small amount of high-quality data for validation, significantly reducing annotation costs. Additionally, recent work on direct on-policy sampling methods (Wu et al., 2024b), which do not need additional SFT data, shows considerable promise. After our final round of improvements, the model-generated responses are already of high quality. Future enhancements can incorporate on-policy preference optimization techniques to further refine the model.

## APPENDIX

## A    ALGORITHM

We provide the overall pipeline of SynPO in Algorithm 1.

---

**Algorithm 1** `Synthetic Preference Optimization (SynPO)`

---

1: **Input**: Initial policy $\pi_{\boldsymbol{\theta}_0}$, validation set $\{(\mathbf{x}_i^*, \mathbf{y}_i^*)\}_{i=0}^n$, keyword list set $\mathcal{K}$, prompt generator $\mathcal{G}$, data filter $\mathcal{F}$, synthetic preference data $\mathcal{D} = \emptyset$

2: **for** $t = 1, 2, \dots$ **do**

3:     Generate $m$ synthetic prompts $\{\mathbf{x}_i\}_{i=1}^m$ with $\mathbf{x}_i \sim \mathcal{G}(\cdot|\mathbf{k}_i)$, where $\mathbf{k}_i$ represents a list of keywords randomly sampled from $\mathcal{K}$.

4:     Train the response improver $\mathcal{R}_t$ from $\boldsymbol{\theta}_0$, $\mathcal{R}_t \leftarrow \arg\min_{\boldsymbol{\theta}} \sum_{i=0}^n \mathcal{L}(\pi_{\boldsymbol{\theta}}(\mathbf{x}_i^*, \mathbf{y}_{(t-1),i}^*), \mathbf{y}_i^*)$, where $\mathbf{y}_{(t-1),i}^* \sim \pi_{\boldsymbol{\theta}_{t-1}}(\cdot|\mathbf{x}_i^*)$ for $i \in \{1, \dots, n\}$.

5:     Generate $\pi_{\boldsymbol{\theta}_{t-1}}$ completions and self-refined completions on the synthetic prompts: $\mathbf{y}_{(t-1),i} \sim \pi_{\boldsymbol{\theta}_{t-1}}(\cdot|\mathbf{x}_i)$, $\overline{\mathbf{y}_{(t-1),i}} \sim \mathcal{R}_t(\cdot|\mathbf{x}_i, \mathbf{y}_{(t-1),i})$ for $i \in \{1, \dots, m\}$.

6:     Filter out invalid refinements and integrate valid data into the synthetic preference dataset:

$$\mathcal{D} \leftarrow \mathcal{D} \cup \left\{ (\mathbf{x}_i, \overline{\mathbf{y}_{(t-1),i}}, \mathbf{y}_{(0),i}) \mid \mathcal{F}(\mathbf{x}_i, \overline{\mathbf{y}_{(t-1),i}}, \mathbf{y}_{(0),i}) = \text{valid}, \ i \in \{1, \dots, m\} \right\}, \mathbf{y}_{(0),i} \sim \pi_{\boldsymbol{\theta}_0}(\cdot|\mathbf{x}_i)$$

7:     Optimize $\pi_{\boldsymbol{\theta}_0}$ using the SimPO (Meng et al., 2024) objective, where $\sigma$ and $\gamma$ are hyperparameters:

$$\boldsymbol{\theta}_t \leftarrow \arg\min_{\boldsymbol{\theta}} \mathbb{E}_{(\mathbf{x}_i, \overline{\mathbf{y}_{(t-1),i}}, \mathbf{y}_{(0),i}) \sim \mathcal{D}} \left[ \log \sigma \left( \frac{\beta}{|\overline{\mathbf{y}_{(t-1),i}}|} \log \pi_\theta(\overline{\mathbf{y}_{(t-1),i}} \mid \mathbf{x}_i) - \frac{\beta}{|\mathbf{y}_{(0),i}|} \log \pi_\theta(\mathbf{y}_{(0),i} \mid \mathbf{x}_i) - \gamma \right) \right]$$

8: **end for**

---

## B    PROMPT FOR RESPONSE-REFINER

The prompt template used for training and inference in the response improver is shown in Figure 7.

---

**Self-Improver Prompt**

You are a smart AI assistant. For a given question-answer pair, improve the answer by correcting errors, bolstering informativeness, aligning with the question, and providing comprehensive detail.

Given Question: {self-generated_question}

Original Answer: {original_model_completion}

Rewritten Answer:

---

Figure 7: Prompt in SynPO for the LLM to act as a response-refiner.

## C    EXPERIMENTAL DETAILS

Here we list additional experimental details for our implementation and experiments.

### C.1    SELF-PROMPT GENERATOR TRAINING

The hyperparameters for self-prompt generator training are detailed below. During SFT for the self-prompt generator, we employ a learning rate of $1.0 \times 10^{-6}$ for Mistral-Base and Llama3-Base, with a batch size of 32, a warm-up ratio of 0.1, and an AdamW optimizer. We set the maximum sequence length to 8,000 and train the model for 3 epochs.

To generate diverse synthetic prompts, we randomly sampled 1 million paragraphs from Refined-Web (Penedo et al., 2023) and randomly selected 3 keywords from each paragraph. This process yields a large keyword list pool containing 1 million keyword lists. These keyword lists serve as the input for the self-prompt generator in each iteration. For each iteration, we generate between 36,000 and 72,000 keyword lists (depending on the filtering ratio at each iteration) and exclude lists containing personal names or stopwords. We use vllm for inference and set the sampling temperature to 0.7.

### C.2    RESPONSE IMPROVER TRAINING

As the model iterates and self-improves, it may produce responses superior to those of the seed data. Our objective is for the response improver to learn from its deficiencies. Therefore, we identify instances where the model output is inferior to the original response using the same scoring model as the filtering stage. This ensures that the response improver only learns positive optimizations or semantic paraphrasing, rather than negative optimizations. Specifically, for a given $\S_i$, if the score difference between the gold standard completion and the model completion exceeds the threshold, we include this data for response improver training. In the Mistral-Base setting, we set the PairRM scoring threshold to 0.20. In the Llama3-Base setting, the ArmoRM-Llama3-8B-v0.1 scoring threshold is set to 0.02.

Since the response improver data are automatically derived from SFT data conversion, the model also learns paraphrasing. Using a more powerful model, such as GPT-4, to create data that introduce only minor improvements for rewriter training is a promising research direction. However, to explore the potential for self-boosting, we did not introduce additional data or stronger models for data construction, resulting in inevitable paraphrasing by the response improver.

During SFT for the response improver, most training parameters are same to the parameters in self-prompt generator training. Some miner differences lie in: we set the max sequence length to 6,000.

### C.3    RESPONSE IMPROVING AND FILTERING SETTING

To produce synthetic preference (chosen and rejected) completions for the $t$-th iteration, we utilize the current policy model to generate a completion and employ the response improver to refine it. We use vllm for inference, with the decoding temperature set at $T = 0.7$.

During synthetic data filtering, we set a threshold 0.20 for PairRM scores and a threshold 0.02 for ArmoRM-Llama3-8B-v0.1 scores. In addition, we filter out all the data that contain over 50% repetition patterns to avoid model collapse on synthetic data. In our experiments, we randomly incorporated 10,000 preference pairs from each iteration to the whole synthetic preference data.

### C.4    OPTIMIZATION

As the parameter $\beta$ is crucial for achieving optimal performance in SimPO (Meng et al., 2024), we individually search the $\beta$ in the range of [2, 4, 6, 8, 10, 12] for each optimization process. We use a fixed $\gamma = 1.6$ for the Mistral-Base model and Llama3-Base.

## C.5 BASELINES

In experiments involving iterative baselines, we control various conditions to ensure fairness. We maintain the same training data size for both iterative baselines and SynPO. We adopt the SimPO loss (Meng et al., 2024) for preference optimization, as it is more effective than DPO (Rafailov et al., 2024). We all use self-generated prompts, which have been shown to be superior to prompts generated by other methods, as validated in Section 2.1 and Section 4.1. All preference construction processes are iterated until performance no longer improves.

Regarding the baseline models trained on UltraFeedback 61k, we straightforwardly adopt the well-trained versions available from the SimPO repository at https://github.com/princeton-nlp/SimPO.

## C.6 DECODING HYPERPARAMETERS

For the AlpacaEval 2 (Dubois et al., 2024) evaluation, we use a sampling-based decoding approach to generate responses. Specifically, we employ vllm for inference, setting the temperature to 0.7 and the maximum tokens to 2048 for both the Mistral-Base and Llama3-Base configurations. All other parameters adhere to the default settings in vllm. As for MT-Bench (Zheng et al., 2024), we adhere to the official decoding setup, which specifies varying sampling temperatures tailored to distinct categories.

## D    ADDITIONAL DETAILS ON SEED DATA ABLATION

In setting 2), 3), and 4), to prevent cases where rejected responses are better than the chosen ones, we filter the preference data using the same method as SynPO, specifically employing ArmoRM-Llama3-8B-v0.1 to select valid preference data. We fix the threshold at 0.02, as our search among {0, 0.1, 0.2} reveal that 0.02 consistently performs the best.

## E    API USAGE

For GPT-4 Turbo, we all use the latest turbo-2024-04-09 API on Azure OpenAI Service    https://learn.microsoft.com/en-us/azure/ai-services/openai/concepts/models#gpt-4-turbo.

## F    PROMPT ANALYSIS

Here we provide the prompt used for prompt topic and intention analysis in Figure 8, along with a more detailed distribution bar plot for different intentions and topics in Figure 9. The topic word list is derived from UltraChat (Ding et al., 2023), while the intention word list was designed by us.

## G    EVALUATION DETAILS

For instruction-following ability evaluation, Table 8 presents the detailed information for three alignment benchmarks we use, including AlpacaEval 2.0, Arena-Hard and MT-Bench. Additionally, we display the radar chart for MT-Bench scores on different prompt types (see Figure 10).

As for general LLM capability evaluation, we provide the few-shot example numbers on Open LLM Leaderboard in Table 9 and a comprehensive comparison of SynPO.

|  | # Instances | Baseline Model | Judge Model | Scoring Type |
|---|---|---|---|---|
| AlpacaEval 2.0 | 805 | GPT-4 Turbo | GPT-4 Turbo | Pairwise comparison |
| Arena-Hard | 500 | GPT-4-0314 | GPT-4 Turbo | Pairwise comparison |
| MT-Bench | 80 | - | GPT-4 Turbo | Single-answer grading |

Table 8: Details for three alignment benchmarks.

---

**Topic and Intention Classification Prompt**

You are a smart AI assistant. Below the '### Given text' section, a user's instruction/query will be provided. Please determine which topic this question belongs to and output ONE most suitable topic from the 'topics' list. Also, output ONE most suitable intention of the user from the 'intentions' list. (Note: try your best to use the words or phrases in the given lists, but if none of them fits, you can output a new one.)

### Given Text

{given_text}

### Topic List topics = ["Technology", "Health and wellness", "Travel and adventure", "Food and drink", "Art and culture", "Science and innovation", "Fashion and style", "Relationships and dating", "Sports and fitness", "Nature and the environment", "Music and entertainment", "Politics and current events", "Education and learning", "Money and finance", "Work and career", "Philosophy and ethics", "History and nostalgia", "Social media and communication", "Creativity and inspiration", "Personal growth and development", "Spirituality and faith", "Pop culture and trends", "Beauty and self-care", "Family and parenting", "Entrepreneurship and business", "Literature and writing", "Gaming and technology", "Mindfulness and meditation", "Diversity and inclusion", "Travel and culture exchange"]

### Intention List intentions = ["Seek advice", "Design help", "Plan something", "Discuss topics", "Analyze something", "Evaluate something", "Search help", "Learn something", "Writing/polishing help", "Quality chat", "Create something", "Fix something", "Compare something", "Transfer something", "Calculate something", "Navigate", "Explore something new", "Play a game", "Install/uninstall help", "Book/cancel help", "Buy/sell suggestions", "Register/enroll help", "Translation help", "Proofreading/editing help", "Mental health advice", "Recommendations", "Troubleshoot help", "Project feedback", "Creative brainstorming", "Time management help", "Organization help", "Public speaking help", "Job application help", "Networking help", "Language learning help", "Technology setup help", "Event coordination help", "Social media management help", "Conflict resolution help", "Sustainable living advice"]

### Output Format Format your chosen topic and intention only as a python dictionary with no extraneous output e.g. "topic": "...", "intention": "...". Each value only contain ONE topic or intention.

Figure 8: Prompt for using GPT-4 Turbo as a intention and topic classifier.

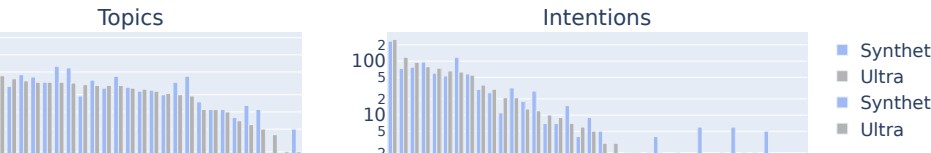

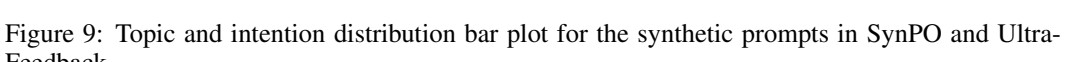

Figure 9: Topic and intention distribution bar plot for the synthetic prompts in SynPO and Ultra-Feedback.

## H   ANALYSIS ON KEYWORDS FOR PROMPT GENERATION

In this section, we systematically investigate the impact of various factors on the quality and diversity of prompts through preliminary experiments on Llama3. We explore the effects of excluding noise keywords (Table 10), the types of keywords used (Table 11), the order of keywords (Table 12), and the number of keywords (Table 13). Our experiments involve training and evaluating the prompt generator with different configurations, measuring average similarity using Sentence-Transformer and assessing prompt quality with GPT-4 Turbo as an LLM-as-a-Judge on a scale from 1 to 10.

Results indicate that the inclusion of noise keywords improves prompt quality as LLMs learn to ignore unrelated words, while using noun phrases enhances diversity and quality. Keyword order has a minimal impact, with randomization slightly enhancing diversity. Additionally, using 3-5 keywords provides the best balance between diversity and quality, with too many keywords degrading prompt quality. These findings collectively guide the implementation of the self-prompt generator.

## I   DATA LEAKAGE ANALYSIS

To ensure the robustness of our evaluation, we conduct a thorough analysis to detect any potential data leakage between our training datasets and the test sets. Specifically, we compare the n-grams

| Task | Arc | HellaSwag | TruthfulQA | MMLU | Winogrande | GSM8k |
|---|---|---|---|---|---|---|
| # Few-shot Examples | 25 | 10 | 0 | 5 | 5 | 5 |
| Metrics | acc_norm | acc_norm | mc2 | acc | acc | acc |

Table 9: Number of few-shot examples in Open LLM Leaderboard evaluation.

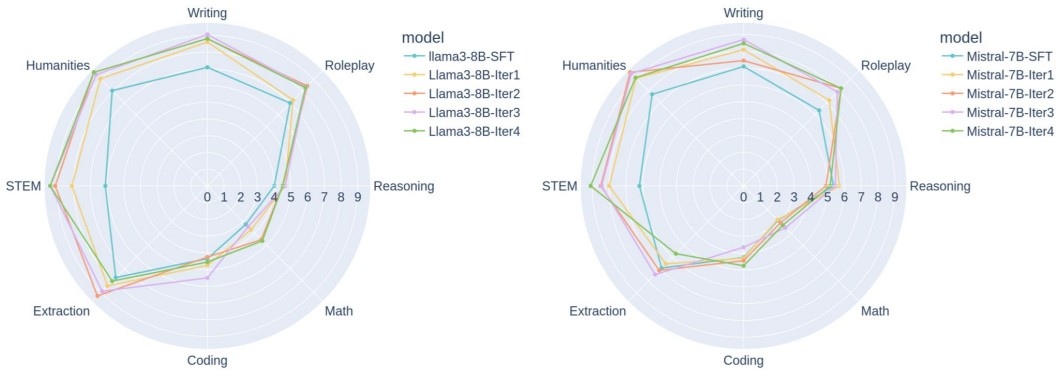

Figure 10: MT-Bench scores on different prompt types. The left radar chart represents results from SynPO on Llama3 and the right comes from Mistral.

of our training datasets (seed SFT data, synthetic preference data, and UltraFeedback for reference) with the n-grams of the test set data to identify any overlaps. If any n-gram from a test data entry appears in our dataset n-grams, that entry is marked as leaked, and we calculate the proportion of leaked data for each test dataset. For datasets with candidate answers, we concatenate the question with candidate answers for analysis; for those without candidate answers, we use only the question. Following Liang et al. (2023), we set the n-gram size to 13.

As shown in Table 14 and Table 15, the overlap between training datasets (UltraFeedback, seed data, and synthetic preference data) and the test sets is very low, indicating that there is no data leakage problem. Interestingly, synthetic preference data generally shows even less overlap with the test sets than other data. These findings enhance the robustness of our research and confirm the integrity of our evaluation process.

## J  EXTENDED EVALUATION ON DIVERSE TASKS

For a comprehensive evaluation across a broader range of tasks, we include ten more tasks from LM Evaluation Harness (see Table 16), including biomedical domain and complex reasoning. We also report the domain-specific results on the AGIEval benchmark (Zhong et al., 2023) (in Table 17), which is a comprehensive benchmark specifically designed to assess foundation models in the context of human-centric standardized exams across diverse domains.

The results indicate that while SynPO is not specifically designed for complex reasoning, it performs competitively across a variety of tasks, particularly in specialized domains. For instance, SynPO 4 iters shows significant improvements over the SFT model in tasks such as EQ-Bench (Paech, 2023). In the AGIEval benchmark, SynPO 4 iters also outperforms the SFT model in each specific domains.

## K  PERFORMANCE GAINS ON TRUTHFULQA

Similar to SimPO Meng et al. (2024), SynPO shows significant enhancement on TruthfulQA Lin et al. (2021). There are two primary reasons for the observed performance gains on the TruthfulQA benchmark. First, there is a strong correlation between preference alignment and TruthfulQA. TruthfulQA evaluates the truthfulness of responses from language models, a goal that closely aligns with preference alignment. Our synthetic preference dataset, which includes instances emphasizing truth-

| Noise Condition | Avg. Similarity | Quality |
|---|---|---|
| No noise | 0.0572 | 7.92 |
| With noise | 0.0574 | 8.99 |

Table 10: Effect of noise keywords.

| Extraction Form | Avg. Similarity | Quality |
|---|---|---|
| Noun Phrases | 0.0574 | 8.99 |
| Verb Phrases | 0.0865 | 8.74 |
| Noun + Verb Phrases | 0.0610 | 8.67 |
| All Phrases | 0.0569 | 8.67 |
| Noun Words | 0.0604 | 8.98 |
| Verb Words | 0.0894 | 8.76 |
| Noun + Verb Words | 0.0725 | 8.52 |
| All Words | 0.0598 | 8.61 |

Table 11: Effect of keyword types.

fulness, enhances the model's ability to accurately interpret context and generate truthful responses. This aligns with findings from Meng et al. (2024). Similarly, we hypothesize that the preference dataset contains instances that emphasize truthfulness, which helps the model better understand the context and generate more truthful responses." Second, we utilize SimPO loss for preference optimization, which has significantly boosted TruthfulQA performance compared to other methods, as detailed in Table 9 of the SimPO paper (Meng et al., 2024).

| Order | Avg. Similarity | Quality |
|---|---|---|
| Keep original order | 0.0605 | 9.01 |
| Random | 0.0574 | 8.99 |

Table 12: Effect of keyword order.

| # of keywords | Avg. Similarity | Quality |
|---|---|---|
| 1 | 0.0621 | 8.71 |
| 3 | 0.0574 | 8.99 |
| 5 | 0.0543 | 8.63 |
| 10 | 0.0541 | 7.08 |

Table 13: Effect of number of keywords.

| Data | Arc | HellaSwag | TQA | MMLU | Winogrande | GSM8k |
|---|---|---|---|---|---|---|
| UltraFeedback (for reference) | 0.00085 | 0.00030 | 0.00122 | 0.00199 | 0.00000 | 0.00531 |
| Seed SFT Data | 0.00085 | 0.00010 | 0.00122 | 0.00036 | 0.00079 | 0.00076 |
| Synthetic Preference Data | 0.00000 | 0.00000 | 0.00122 | 0.00064 | 0.00000 | 0.00000 |

Table 14: Proportion of leaked data for various test datasets when compared with UltraFeedback, Seed SFT Data, and Synthetic Preference Data.

| Data | AlpacaEval 2.0 | Areana-Hard | MT-Bench |
|---|---|---|---|
| UltraFeedback | 0.00248 | 0.00600 | 0.01250 |
| Seed SFT Data | 0.00373 | 0.01200 | 0.01250 |
| Synthetic Preference Data | 0.00124 | 0.00800 | 0.00000 |

Table 15: Proportion of leaked data for additional test datasets when compared with UltraFeedback, Seed SFT Data, and Synthetic Preference Data.

| Model | LogiQA (Liu et al., 2020) | MMLU-Pro (Wang et al., 2024b) | SiQA (Sap et al., 2019) | QA4MRE (Peñas et al., 2013) | NQ (Kwiatkowski et al., 2019) |
|---|---|---|---|---|---|
| SFT | 30.88 | 36.12 | 46.21 | 46.13 | 11.55 |
| SynPO 4 iters | 31.95 | 37.28 | 49.18 | 49.30 | 12.80 |

| Model | PubMedQA (Jin et al., 2019) | RACE (Lai et al., 2017) | SWAG (Zellers et al., 2018) | EQ-Bench (Paech, 2023) | Story Cloze (Mostafazadeh et al., 2016) |
|---|---|---|---|---|---|
| SFT | 73.20 | 44.59 | 76.91 | 46.79 | 10.55 |
| SynPO 4 iters | 75.44 | 46.70 | 77.19 | 55.82 | 12.80 |

Table 16: Evaluation results on 10 additional LM Evaluation Harness tasks, using Llama3.

| Model | History | Biology | Chemistry | Physics | MathQA |
|---|---|---|---|---|---|
| Llama3-SFT | 38.29 | 34.29 | 24.64 | 32.00 | 24.22 |
| SynPO 4 iters | 45.53 | 39.05 | 31.88 | 35.00 | 28.49 |

Table 17: Evaluation results on AGIEval for different domains, using Llama3.

