# OpenReview forum: "Self-Boosting Large Language Models with  Synthetic Preference Data"
_ICLR.cc/2025/Conference — ICLR 2025 Poster_

### Official Review · Reviewer_d9yw · 2024-10-26

**Soundness:** 3
**Presentation:** 2
**Contribution:** 2
**Rating:** 6
**Confidence:** 3

**Summary:**

The authors introduce SynPO (Self-Boosting Preference Optimization), a framework that enables LLMs to autonomously generate synthetic data and enhance their performance through iterative self-boosting.  SynPO leverages a self-prompt generator and response improver to collaboratively produce high-quality synthetic preference data for training. This process allows the model to learn responses aligned with preferences, leading to performance improvements on benchmarks like the Open LLM Leaderboard, AlpacaEval 2.0, and Arena-Hard.

**Strengths:**

1. The authors contribute by demonstrating infinite generation potential through experiments that examine performance changes relative to the number of synthetic datasets. They also clarify the limitations by distinguishing this capacity from the sustained utility of such datasets over time.

2. The contribution to diversity within the keyword list is effectively demonstrated, with supporting experiments that validate this aspect.

3. Previous research has rarely considered multi-turn interactions; however, the authors establish the effectiveness of their proposed method under multi-turn dialogue conditions.

4. The self-boosting approach employed in the optimization process is appropriate and well-suited for enhancing model performance.

**Weaknesses:**

1. An ablation study on the effect of excluding noise keywords is necessary for Lines 156–157.

2. Analysis of the types of keywords is insufficient. The authors justify random sampling based on diversity, yet the study lacks analysis on which extraction forms might be more beneficial, or order affects outcomes.

3. It would be interesting to see experiments that vary the number of keywords or increase them significantly.

4. The most concerning issue is the weak validation of out-of-distribution data. Since the seed dataset reflects OpenAI GPT’s preferences, as do Alpaca 2.0 and Arena-Hard, this likely influences the performance shown in Table 2. Additionally, while Table 4 shows considerable improvement on TQA included in the UltraFeedback dataset, effects are minimal on other datasets, suggesting that SynPO does not fully resolve in-distribution bias issues. There are also doubts about whether this approach could be more efficient or effective than methods in other studies applying CoT rather than simple SFT.

**Questions:**

1. References are needed for studies on a word-to-text task [1] and similar augmentation research where word sets are extracted from sentences in existing datasets, combined to create new synthetic datasets, and enhanced in quality using contrastive learning [2].

    [1] *Lin, B. Y., Zhou, W., Shen, M., Zhou, P., Bhagavatula, C., Choi, Y., & Ren, X. (2020, November). CommonGen: A Constrained Text Generation Challenge for Generative Commonsense Reasoning. In Findings of the Association for Computational Linguistics: EMNLP 2020 (pp. 1823-1840).*

    [2] *Seo, J., Moon, H., Lee, J., Eo, S., Park, C., & Lim, H. S. (2023, December). CHEF in the Language Kitchen: A Generative Data Augmentation Leveraging Korean Morpheme Ingredients. In Proceedings of the 2023 Conference on Empirical Methods in Natural Language Processing (pp. 6014-6029).*

---

> ### Author Response · Authors · 2024-11-20
> **Reply to Reviewer d9yw (Part 1/3 )**
>
> We sincerely thank Reviewer d9yw for your review and are grateful for the time you spent on our submission. We're pleased you find our method effective and well-validated. Below, we provide a point-by-point rebuttal to clarify your concerns.
>
> **Q1: Ablation study on the effect of excluding noise keywords**
>
> As suggested, we conduct experiments on the effect of excluding noise keywords. We train and evaluate the Llama3 prompt generator with three random keywords, either including or excluding one noise keyword. We calculate the average similarity across the generated prompts using SentenceTransformer, as mentioned in Section 2.2, and employ GPT-4 Turbo for LLM-as-a-Judge evaluation of prompt quality (on a scale of 1 to 10, where 1 represents a prompt that is very unrealistic, unnatural, or unanswerable, and 10 represents a prompt that is very reasonable, realistic, and answerable).
>
> |Noise Condition | Avg Similarity | Quality |
> |-----------------|----------------|---------|
> | no noise        | 0.0572         | 7.92   |
> | w noise         | 0.0574          | 8.99   |
> |
>
> *( Evaluation run on 1k data for each setting, each example contains 3 keywords )*
>
> The results show that the inclusion of noise keywords improves the quality of the self-generated prompts (as LLMs learn to ignore unrelated words), but even without noise keywords, the self-prompt generator can still generate relatively high-quality prompts.
>
>
>
> **Q2: Analysis of the types of keywords**
>
> **1. Extraction Form**
>
> Currently, our extraction form employs a rule-based random selection. Specifically, we use the NLTK toolkit to filter out stop words, extract all noun phrases from the sentences, and remove any preceding articles. We then randomly sample from these phrases. The decision to use noun phrases is based on following preliminary experiments:
>
> | Extraction Form | Avg. Similarity | Quality |
> | :---- | :---- | :---- |
> | Noun Phrases | 0.0574 | 8.99 |
> | Verb Phrases | 0.0865 | 8.74 |
> | Noun \+ Verb Phrases | 0.0610 | 8.67 |
> | All Phrases | 0.0569 | 8.67 |
> | Noun Words | 0.0604 | 8.98 |
> | Verb Words | 0.0894 | 8.76 |
> | Noun \+ Verb Words | 0.0725 | 8.52 |
> | All Words | 0.0598 | 8.61 |
> |
>
> *(Note: "Words" refer to single-word keywords, while "Phrases" refer to each keyword can be a phrase comprising 1 to 3 words.)*
>
> As shown, **using phrases rather than limiting keywords to single words generally results in better diversity and quality**. This improvement may be due to phrases incorporating more inductive bias from the pretraining data for prompt generation. Specifically, while random sampling from all phrases or words without filtering is acceptable, it is not as effective as **random sampling from nouns**, as nouns contain the most important information needed to diversify a sentence.
>
> Beyond these evaluation results, we have also manually observed the impact of different extraction forms and found that using phrases as keywords generally resulted in higher quality prompts. Therefore, our experiments utilize randomly sampled noun phrases. We will include these preliminary experiments, which support our choice of keyword extraction methods, in the Appendix to provide further insights into self-prompt generator training.
>
> **2. Order**
>
> Whether order influences is a very insightful point. In our current approach, we randomize keyword order. To systematically examine this, we compared results from maintaining the original keyword order (also in the pretraining data) versus randomizing it. The results are shown in the table below.
>
> | Order | avg similarity | Quality |
> | :---- | :---- | :---- |
> | keep original order | 0.0605 | 9.01 |
> | random | 0.0574 | 8.99 |
> |
>
> *( Evaluation run on 1k data for each setting, each example contains 3 keywords )*
>
>
>
>
> The results indicate that **keyword order has a minimal impact on prompt generation**. Randomization enhances diversity, while preserving the original order retains some semantic context from the pretraining data, yielding slightly higher quality prompts. However, this effect is negligible.
>
>
> **Q3: Vary the number of keywords**
> Thanks for the great suggestion. As suggested, we have added experiments that vary the number of keywords, and the results are presented in the following table. The experimental findings indicate that for training the self-prompt generator, **using 3-5 keywords strikes a good balance between the diversity and quality of the generated prompts**. Including too many keywords can be detrimental to the quality of the generated prompts.
>
> | \# of keywords | Avg. Similarity | Quality |  |
> | :---- | :---- | :---- | :---- |
> | 1 | 0.0621 | 8.71 |  |
> | 3 | 0.0574 | 8.99 |  |
> | 5 | 0.0543 | 8.63 |  |
> | 10 | 0.0541 | 7.08 |  |
> |
>
> *( Evaluation run on 1k data for each setting. )*

---

> > ### Author Response · Authors · 2024-11-20
> > **Reply to Reviewer d9yw (Part 2/3 )**
> >
> > **Q4: Validation of out-of-distribution data**
> >
> > We would like to highlight that for evaluation, in addition to instruction-following tasks, we have assessed the model's performance on diverse downstream tasks that are not in-distribution data, **including six varied LLM Harness tasks and six Open LLM Leaderboard tasks**.
> >
> > To address potential concerns, we provide more diverse evaluation results on additional tasks, including various domains in AGIEval benchmarks and 10 downstream tasks spanning a wide range of evaluation data.
> >
> > **A. AGIEval Domains**
> > | Model       | History | Biology | Chemistry | Physics | MathQA |
> > |---------------|---------|---------|-----------|---------|--------|
> > | Llama3-SFT    | 38.29   | 34.29   | 24.64     | 32.00   | 24.22  |
> > | SynPO 4 iters  | 45.53   | 39.05   | 31.88     | 35.00   | 28.49  |
> > |
> >
> > *(Evaluation results on AGIEval for different domains, using Llama3)*
> >
> >
> > **B. More Diverse Tasks**
> >
> > | Model      | LogiQA |MMLU-Pro| SiQA | QA4MRE | NQ Open | PubMedQA | RACE  | SWAG  | EQ-Bench | Story Cloze |
> > |---------------|--------|--------|------------|--------|---------|----------|-------|-------|----------|-------------|
> > | SFT           | 30.88 |36.12 | 46.21      | 46.13  | 11.55   | 73.20    | 44.59 | 76.91 | 46.79    | 10.55       |
> > | SynPO 4 iters | 31.95|37.28  | 49.18      | 49.30  | 12.80   | 75.44    | 46.70 | 77.19 | 55.82    | 12.80       |
> > |
> >
> > *(Evaluation results on 10 additional LM Evaluation Harness tasks for reasoning, using Llama3)*
> >
> > **Specifics for AGIEval:**
> >
> > AGIEval is a comprehensive benchmark specifically designed to assess foundation models in the context of human-centric standardized exams across diverse domains, such as college entrance exams, law school admission tests, math competitions, and lawyer qualification tests.
> >
> > **Other Task Specifics:**
> >
> > - LogiQA: Logical reasoning tasks requiring advanced inference and deduction.
> >
> > - MMLU-Pro:  A refined set of MMLU, integrating more challenging, reasoning-focused questions and expanding the choice set from four to ten options.
> >
> > - SiQA: Social Interaction Question Answering to evaluate common sense and social reasoning.
> >
> > - QA4MRE: Question Answering for Machine Reading Evaluation, assessing comprehension and reasoning.
> >
> > - NQ Open: Open domain question answering tasks based on the Natural Questions dataset.
> >
> > - PubMedQA: Question answering tasks based on PubMed research articles for biomedical understanding.
> >
> > - RACE: Reading comprehension assessment tasks based on English exams in China.
> >
> > - SWAG: Situations With Adversarial Generations, predicting the next event in videos.
> >
> > - EQ-Bench: Tasks focused on equality and ethics in question answering and decision-making.
> >
> > - Story Cloze: Tasks to predict story endings, focusing on narrative logic and coherence.
> >
> >
> >
> > **Q5: Comparison with COT methods**
> >
> > - Our approach and CoT methods are complementary. SynPO addresses how to iteratively generate preference data for continuous training of the model given a small amount of seed SFT data, while CoT focuses on enhancing the reasoning process and rationale in the generated SFT data.
> >
> > - To further illustrate the complementarity between our method and CoT, we conducted the following experiment:
> >
> > | Method | LC (%) | WR (%) |
> > | :---- | :---- | :---- |
> > | Seed SFT | 20.1 | 19.7 |
> > | Seed SFT (COT)  | 21.5 | 22.4 |
> > | SynPO  | 32.1 | 33.6 |
> > | SynPO (COT) | 32.3 | 35.4 |
> > |
> >
> > The experiment was conducted under the same settings as Table 7 in our paper. Following Mukherjee et al. \[1\], the results for SFT (CoT) and SynPO (CoT) were obtained by directly applying CoT to seed data SFT and iteratively training with the SynPO method under identical conditions. Specifically, the CoT method involved randomly selecting one of the system messages for CoT from Mukherjee et al.'s approach during both the seed data creation and model response generation stages. This resulted in seed data and self-generated preference data with more extensive reasoning processes. The results indicate that incorporating CoT significantly improves the win rate on AlpacaEval 2.0, while having a minimal impact on the length-controlled win rate.

---

> ### Author Response · Authors · 2024-11-20
> **Reply to Reviewer d9yw (Part 3/3 )**
>
> **Q6: Complementary references**
> Thanks for pointing out the related literature\!  Lin et al.'s work \[1\] shares a similar spirit with our prompt generator in generating sentences from keywords, but they focus on explicitly testing machines' commonsense reasoning ability by generating a coherent sentence with a given set of common concepts. Seo et al.'s paper \[2\] leverages Korean morphological variations to create synthetic data, which is then enhanced in quality using contrastive learning. We will add discussion on these related and insightful works in Section 5 in our final revision.
>
>
> Thank you very much for the constructive comments, which really help us further improve our work. We hope our answers have addressed your concerns. If you have any further questions, we are happy to address them.
>
> **Reference**
> *\[1\] Mukherjee, Subhabrata, et al. "Orca: Progressive learning from complex explanation traces of gpt-4." arXiv preprint arXiv:2306.02707 (2023).*
> *\[2\] Lin, B. Y., Zhou, W., Shen, M., Zhou, P., Bhagavatula, C., Choi, Y., & Ren, X. (2020, November). CommonGen: A Constrained Text Generation Challenge for Generative Commonsense Reasoning. In Findings of the Association for Computational Linguistics: EMNLP 2020 (pp. 1823-1840).*
> *\[3\] Seo, J., Moon, H., Lee, J., Eo, S., Park, C., & Lim, H. S. (2023, December). CHEF in the Language Kitchen: A Generative Data Augmentation Leveraging Korean Morpheme Ingredients. In Proceedings of the 2023 Conference on Empirical Methods in Natural Language Processing (pp. 6014-6029).*

---

> > ### Author Response · Authors · 2024-11-24
> > **Further comments and discussions will be appreciated!**
> >
> > Dear Reviewer d9yw,
> >
> > Thank you for your valuable time to review our work and constructive feedback. We posted our response to your comments four days ago, and we wonder if you could kindly share some of your thoughts so we can keep the discussion rolling to address your concern if there are any.
> >
> > In the previous response,
> >
> > 1. As suggested, we added discussion on the mentioned works in Section 5 in the revised version.
> >
> > 2. We discussed and added our preliminary experiments on keywords analysis  in the Appendix H which support our choice of keyword extraction methods, to provide further insights into self-prompt generator training.
> >
> > 3. We conduct additional experiments to justify the complementarity of our method with CoT.
> >
> > 4. We added experiments to provide a more comprehensive evaluation on more broad tasks and out-of-distribution data, we added ten additional tasks (including reasoning tasks and domain specific tasks) and reported the results in domain-specific scores on AGIEval in Appendix J.
> >
> >
> > We would appreciate it if you could kindly take a look at both the revision and our response to your comments. If you have any further questions, we are happy to discuss them\!
> >
> > Best regards,
> >
> > Authors

---

> > > ### Author Response · Authors · 2024-11-25
> > > **Follow up to reveiwer d9yw**
> > >
> > > Dear Reveiwer d9yw,
> > >
> > > We would like to thank you again for your detailed reviews. We have updated our draft and added replies to your Cons with our latest experimental results.
> > >
> > > Since the rebuttal deadline is approaching soon, a lot of papers have finished the discussion. Given that your current score is 5, we would appreciate it if you could let us know if our responses have addressed your concerns satisfactorily. If your concerns have not been resolved, could you please let us know about it so that we have the opportunity to respond before the deadline?
> > >
> > > We would be happy to have any follow-up discussions or address any additional concerns.
> > >
> > > Thanks very much! Looking forward to your reply.
> > >
> > > Best,
> > >
> > > Authors

---

> > > > ### Comment · Reviewer_d9yw · 2024-11-25
> > > >
> > > > I believe the updates provided by the authors will further enhance the clarity and strength of this paper. Thank you for your response, and I will adjust the score accordingly.

---

> > > > > ### Author Response · Authors · 2024-11-26
> > > > > **Response to Reviewer d9yw**
> > > > >
> > > > > Thanks a lot for your positive feedback and for raising your score. We're really glad to hear that our revisions have improved the paper. If you have any further questions, we are happy to address them.

---

### Official Review · Reviewer_Ftfv · 2024-10-31

**Soundness:** 3
**Presentation:** 4
**Contribution:** 3
**Rating:** 6
**Confidence:** 4

**Summary:**

The paper introduces SynPO, a self-boosting framework that improves LLMs using synthetic preference data rather than human annotations. The key innovation is a two-part process: (1) A self-prompt generator creates diverse prompts using just three keywords, and (2) A response improver that refines model outputs. The refined outputs are used as "chosen" responses and the original outputs as "rejected" responses for preference learning. Testing on Llama3-8B and Mistral-7B shows significant improvements in instruction following (~22-30% win rate improvements on benchmarks) and general performance (3-5% increase on Open LLM leaderboard) after four iterations.

**Strengths:**

Novel and Practical Solution:
Addresses a critical challenge in LLM development - the scarcity of high-quality preference data
Provides a scalable way to generate training data without relying on human annotations
Simple but effective keyword-based prompt generation strategy

**Weaknesses:**

Limited Task Scope:
Primary evaluation focuses on instruction-following tasks with limited testing on complex reasoning tasks or specialized domains
Some tasks show performance degradation (e.g., Mistral's performance on GSM8K)


Unexplained Performance Gaps:
The dramatic improvement over SFT baseline (e.g., from 6.6% to 34.0% win rate) needs more analysis
Limited ablation studies on why the synthetic data works so much better

**Questions:**

See weaknesses

---

> ### Author Response · Authors · 2024-11-20
> **Reply to Reviewer Ftfv (Part 1/2 )**
>
> We sincerely thank Reviewer Ftfv for your review and are grateful for the time you spent on our submission. We are glad for the acknowledgment that our approach is novel, effective and practical to tackling a critical challenge in LLM development. Below we would like to give detailed responses to each of your comments.
>
> **Q1: Limited task scope**
>
> Except for instruction-following tasks, we have also evaluated the model performance on **six diverse LM Evaluation Harness tasks** (Table 5, including ARC challenge for reasoning and GSM8k for math domain), as well as **six Open LLM Leaderboard  tasks** (Table 4, including PROST for reasoning and MathQA for math domain).
>
>
> We would like to note that SynPO is not designed for complex reasoning but we agree that comprehensive evaluation across a broader range of tasks would strengthen our findings. To address potential concerns regarding task scope, we have extended our evaluation to include more complex reasoning tasks and specialized domains as suggested.
>
>
> - For complex reasoning tasks, we have evaluated the model performance on **seven additional reasoning-related tasks** from LM Evaluation Harness. The results are presented below:
>
>
> | Model        | LogiQA | LogiQA 2.0 | MMLU-Pro| SiQA | QA4MRE | PIQA | NQ Open |
> |---------------|--------|--------|---------|------------|--------|-------|---------|
> | Llama3-SFT           | 30.88  | 36.39  |36.12 | 46.21      | 46.13  | 81.61 | 11.55   |
> | SynPO 4 iters  | 31.95  | 37.40  |37.28  | 49.18      | 49.30  | 81.83 | 12.80   |
> |
>
> *(Evaluation results on 7 LM Evaluation Harness tasks for reasoning, using Llama3)*
>
>
> - For specialized domains, we evaluated model performance on **four additional tasks from diverse domains** and reported the **domain-specific results on the AGIEval benchmark**:
>
>
> | Model     | PubMedQA | RACE  | SWAG  | EQ-Bench |
> |---------------|----------|-------|-------|----------|
> | Llama3-SFT    | 73.20    | 44.59 | 76.91 | 46.79    |
> | SynPO 4 iters  | 75.44    | 46.70 | 77.19 | 55.82    |
> |
>
> *(Evaluation results on 4 LM Evaluation Harness tasks for different domains, using Llama3)*
>
>
> | Model      | History | Biology | Chemistry | Physics | MathQA |
> |---------------|---------|---------|-----------|---------|--------|
> | Llama3-SFT    | 38.29   | 34.29   | 24.64     | 32.00   | 24.22  |
> | SynPO 4 iters  | 45.53   | 39.05   | 31.88     | 35.00   | 28.49  |
> |
>
> *(Evaluation results on AGIEval for different domains, using Llama3)*
>
>
> **Specifics for AGIEval:**
>
> AGIEval is a comprehensive benchmark specifically designed to assess foundation models in the context of human-centric standardized exams across diverse domains, such as college entrance exams, law school admission tests, math competitions, and lawyer qualification tests.
>
>
> **Other Task Specifics:**
>
> - LogiQA: Logical reasoning tasks requiring advanced inference and deduction.
>
> - LogiQA 2.0: Large-scale logical reasoning dataset adapted from the Chinese Civil Service Examination.
>
> - MMLU-Pro:  A refined set of MMLU, integrating more challenging, reasoning-focused questions and expanding the choice set from four to ten options.
>
> - SiQA: Social Interaction Question Answering to evaluate common sense and social reasoning.
>
> - QA4MRE: Question Answering for Machine Reading Evaluation, assessing comprehension and reasoning.
>
> - PIQA: Physical Interaction Question Answering tasks to test physical commonsense reasoning.
>
> - NQ Open: Open domain question answering tasks based on the Natural Questions dataset.
>
> - PubMedQA: Question answering tasks based on PubMed research articles for biomedical understanding.
>
> - RACE: Reading comprehension assessment tasks based on English exams in China.
>
> - SWAG: Situations With Adversarial Generations, predicting the next event in videos.
>
> - EQ-Bench: Tasks focused on equality and ethics in question answering and decision-making.
>
> **Q2: Unexplained performance gaps**
> We'd like to note that the scoring method used in AlpacaEval is based on **pair-wise comparison** (as shown in Table 8). This approach reflects the relative performance improvement over the baseline model, which can result in a larger numerical scale of improvement (e.g., SimPO training on UltraFeedback improves AlpacaEval score from 6.2 to 22.0 \[1\]). When we consider absolute score changes on direct evaluation benchmarks (such as the MT-Bench results in Table 3\) or accuracy-based benchmarks (like the Open LLM Leaderboard results in Table 4), the improvements are more moderate but still consistently positive.

---

> ### Author Response · Authors · 2024-11-20
> **Reply to Reviewer Ftfv (Part 2/2 )**
>
> **Q3: Regarding why synthetic data works better and ablation studies**
>
> Regarding why synthetic data performs better and related ablation studies, we would like to clarify that our comparison with the SFT baseline does not involve contrasting synthetic data with real data. With limited SFT data as seed data, our goal is to construct effective preference data to continuously improve LLMs' instruction-following and end-task capability. The SFT baseline serves as the initial checkpoint for further self-boosting.
>
> As for the reasons why synthetic data works well, there are several main reasons:
>
> 1. Compared to human-collected preference data (whose format is also preference pairs), SynPO is an iterative and nearly on-policy process. At each iteration, the current model state is considered for constructing synthetic preference data. As validated by Chen et al.\[2\] and Yuan et al.\[3\], **on-policy preference data are usually more effective than static data**.
> 2. Compared to continuously training on seed SFT data (as shown in our ablation study in Section 4.2, Table 7), in scenarios where only a limited amount of seed SFT data is available, the number of SFT data is limited and **the format of SFT is not as effective for alignment as preference pair format data**. Preference pairs can lead to more effective learning of human preferences than SFT data, which may overfit to specific examples\[2\]. Our approach of self-synthesizing effective preference pairs allows for iterative improvements in model performance.
>
> In Table 6, we further compare the effects of synthetic prompts and manually collected prompts of the same scale. The results show that **synthetic prompts perform slightly better due to two main reasons**:
> 1. Synthetic data can **better control for diversity** (as shown in Figures 4 and 5).
> 2. Our method allows the model to **gradually self-learn the distribution gap** during the self-improvement training process, enabling it to address specific shortcomings at each stage rather than repeatedly learning from a fixed distribution as with real data.
>
>
> Overall, we greatly appreciate your efforts for your thoughtful and insightful comments on our paper. We hope our answers have addressed your concerns.
>
>
> **Reference**
>
> *\[1\] Meng, Yu, Mengzhou Xia, and Danqi Chen. "Simpo: Simple preference optimization with a reference-free reward." arXiv preprint arXiv:2405.14734 (2024).*
>
> *\[2\] Chen Z, Deng Y, Yuan H, et al. Self-play fine-tuning converts weak language models to strong language models\[J\]. arXiv preprint arXiv:2401.01335, 2024\.*
>
> *\[3\] Yuan, Weizhe, et al. "Self-rewarding language models." arXiv preprint arXiv:2401.10020 (2024).*
>
> *\[4\] Ouyang L, Wu J, Jiang X, et al. Training language models to follow instructions with human feedback\[J\]. Advances in neural information processing systems, 2022, 35: 27730-27744.*

---

> > ### Author Response · Authors · 2024-11-24
> > **Further comments and discussions will be appreciated!**
> >
> > Dear Reviewer Ftfv,
> >
> > Thank you for your valuable time to review our work and constructive feedback. We posted our response to your comments four days ago, and we wonder if you could kindly share some of your thoughts so we can keep the discussion rolling to address your concern if there are any.
> >
> > In the previous response,
> >
> > 1. As suggested, we provided a more comprehensive evaluation on more broad tasks and domains, we added ten additional tasks (including reasoning tasks and domain specific tasks) and reported the results in domain-specific scores on AGIEval in Appendix J.
> >
> > 2. We explained that the scoring method in AlpacaEval is based on pair-wise comparison, which can result in larger numerical improvements (with related results in SimPO paper for reference).
> >
> > 3. We clarified that our comparison with the SFT baseline does not involve contrasting synthetic data with real data. We further explained why synthetic data performs better at the prompt level, because synthetic prompts perform slightly better due to better diversity control and gradual self-learning of the distribution gap.
> >
> > We would appreciate it if you could kindly take a look at both the revision and our response to your comments. If you have any further questions, we are happy to discuss them!
> >
> > Best regards,
> >
> > Authors

---

> > > ### Author Response · Authors · 2024-11-27
> > >
> > > Dear Reviewer Ftfv,
> > >
> > > We would like to thank you again for your detailed reviews. We have updated our draft and added replies to your questions with our latest experimental results.
> > >
> > > Since the rebuttal deadline is approaching soon, we would appreciate it if you could let us know if our responses have addressed your concerns satisfactorily. If so, we would be grateful if you could consider raising your score. We would be happy to have any follow-up discussions or address any additional concerns.
> > >
> > > Thanks very much! Looking forward to your reply.
> > >
> > > Best,
> > >
> > > Authors

---

### Official Review · Reviewer_LmCF · 2024-11-01

**Soundness:** 3
**Presentation:** 3
**Contribution:** 3
**Rating:** 8
**Confidence:** 2

**Summary:**

This paper proposes a framework called SynPO that leverages synthetic preference data for human alignment. The framework starts with the training of a self-prompt generator to create large-scale synthetic prompts. This prompt-generator is trained from the LLM itself, starting from some initial SFT data, and can generate prompts based on keywords. In an iterative process, these prompts are then given to the model to be trained, and passed through a response improver, which is retrained each round to reduce the gap between policy model outputs and gold standard responses. The authors perform experiments with Mistral-Base 7B and Llama3-8B, starting from versions that have undergone SFT supervision on a chat-dataset. They show improvements on three alignment benchmarks: MT-Bench, Arena Hard and AlpacaEval 2.0. They also show an overall improvement on standard benchmarks including Arc, HellaSwag, TQA and MMLU.

**Strengths:**

The paper does quite extensive experimentation, investigate the diversity of the prompts and do ablations to understand the impact of various parts of their pipeline. The topic will be well-received I believe given the recent attention for alignment and the cost of generating human preference data.

**Weaknesses:**

In the introduction, it is not entirely clear to me what model is what. E.g. it is mentioned that a self-prompt generator is trained to create synthetic prompts, and after that it is mentioned that model-generated responses are used as rejected candidates and a response improver is used to improve the response. It is also mentioned that a small amount of SFT data is used to steer the generation of synthetic preference data. But what is what? Which model is used with the synthetic preference data? What is the "response"? Is that coming from the self-prompt generator? Are the self-prompt generator and the response improver the same model? Some of this is cleared up later, but the reader does not know that while trying to comprehend that initial paragraph.

Small point: I don't think it is entirely accurate to write that you use Mistral-base and Llama 3 base model, because those models did not undergo SFT. The models stem from these models (as does the Llama 3 8B instruct model), but they are not those base models.

Lastly, I am a bit confused about the improvement on the standard benchmarks. Many of these benchmarks are things learned in pretraining (e.g. knowledge). The fact that some of these scores go up, especially on knowledge benchmarks, is a bit suspicious because knowledge is something that can evidently not be learned from just self-improvement loops. It makes me wonder if the improvement is just due to the input coming from GPT-4. I would ask if you considered a basilen where you do not update the policy model in between, but just do multiple rounds trying to squeeze out more of the GPT generated data without doing iterations with the model itself, or just do more SFT with the initial data, but I think that those are the ablations presented in 4.2 and the difference seems quite large.

**Questions:**

The fact that some of these scores go up, especially on knowledge benchmarks, is a bit suspicious because knowledge is something that can evidently not be learned from just self-improvement loops. It makes me wonder to what extent the improvement is just coming from the GPT-4 data. Your experiments in 4.2 seem to refute that idea (though you don't look at the benchmarks there I believe), leaving me a bit puzzled. Do you have any other explanation for why the model would improve on something it shouldn't improve on?

**Details Of Ethics Concerns:**

I don't think the OpenAI terms allow the use of outputs of their models to train other models.

---

> ### Author Response · Authors · 2024-11-21
> **Reply to Reviewer LmCF**
>
> We really appreciate your effort for reviewing our paper and your acknowledgement of our paper’s contribution.  We are also glad for the acknowledgment that the problem our experiments are extensive and the topic will be well-received. Below, we would like to give detailed responses to each of your comments.
>
> **Q1: Clarity of the initial  paragraph**
> Thanks for pointing out the potential confusion. We will clarify them point by point below and update our expressions in paragraph 3 in our introduction for better understanding.
>
> * The initial model, which undergoes preference optimization, is first trained to be a prompt generator and is used with the synthetic preference data.
> * The "response" refers to the outputs generated by the initial model when given synthetic prompts. Only the prompts are generated by the self-prompt generator; the responses are the LLM's completions of these prompts.
> * The self-prompt generator and the response improver are trained from the same base model.
>
> Also, we present the SynPO algorithm in Appendix A for a more higher formulated explanation for reference.
> Thank you very much for the constructive comments again, which really help us further improve our clarity.
>
> **Q2: Small point for model name**
>
> Thanks for pointing out this\! Yes, we agree that using ‘-base’ may bring some confusions, and we have replaced the model names from Mistral-base, Llama3-base to Mistral-base-SFT and Llama3-base-SFT, respectively. We’ll include this revision in our latest version.
>
> **Q3: Confusion for improvement on things learned in pretraining**
>
> As shown in Table 5, SynPO primarily enhances performance in commonsense reasoning tasks (such as ARC and HellaSwag) and model honesty (TruthfulQA). For knowledge benchmarks like MMLU, the focus is more on maintaining performance rather than improving it.
>
> Regarding the observed improvements in certain knowledge-related benchmarks (e.g., OBQA), there are two main explanations:
>
> 1. **Alignment Benefits**: We agree that the knowledge contained within the model is acquired during the pretraining phase. However, instruction-tuning and alignment can enable the model to better induce and articulate this knowledge in its responses\[1,2,3\]. For example, Self-Rewarding\[1\] has improved Llama2's performance on Natural Questions.
> 2. **Self-Refinement Gains:** Our method involves a self-refinement process, which has been shown to be effective in various tasks \[4, 5\]. In most scenarios, verification and refinement can be simpler than direct generation, allowing the model to correct inaccuracies and improve its performance \[6,7\]. This process of pre- and post-refinement helps the model enhance its capabilities on these tasks.
>
> Yes, our ablation study detailed in Section 4.2 examines the impact of seed SFT data. The results, presented as "Seed SFT results" in Table 7, demonstrate that Llama3 reaches optimal performance after the second epoch. Additional training epochs did not yield improvements and, at times, even diminished performance. This confirms that the self-refinement process effectively enhances the model's ability to follow instructions and perform tasks, rather than simply benefiting from GPT-4 input.
>
> Again, we sincerely thank you for the constructive comments and positive feedback, which really help us further improve our work. If you have any further questions, we are happy to address them.
>
> **Reference:**
>
> *\[1\] Yuan, Weizhe, et al. "Self-rewarding language models." arXiv preprint arXiv:2401.10020 (2024).*
>
> *\[2\] Chen Z, Deng Y, Yuan H, et al. Self-play fine-tuning converts weak language models to strong language models\[J\]. arXiv preprint arXiv:2401.01335, 2024\.*
>
> *\[3\]  Yin, Yueqin, et al. "Self-Augmented Preference Optimization: Off-Policy Paradigms for Language Model Alignment." arXiv preprint arXiv:2405.20830 (2024).*
>
> *\[4\] Pan, Liangming, et al. "Automatically correcting large language models: Surveying the landscape of diverse self-correction strategies." arXiv preprint arXiv:2308.03188 (2023).*
>
> *\[5\] Weng, Yixuan, et al. "Large language models are better reasoners with self-verification." arXiv preprint arXiv:2212.09561 (2022).*
>
> *\[6\] Kamoi R, Zhang Y, Zhang N, et al. When can llms actually correct their own mistakes? a critical survey of self-correction of llms\[J\]. Transactions of the Association for Computational Linguistics, 2024, 12: 1417-1440.*
>
> *\[7\] Madaan, Aman, et al. "Self-refine: Iterative refinement with self-feedback." Advances in Neural Information Processing Systems 36 (2024).*

---

> > ### Comment · Reviewer_LmCF · 2024-11-25
> >
> > Thank you for the clarifications :)

---

### Official Review · Reviewer_EKkD · 2024-11-04

**Soundness:** 3
**Presentation:** 4
**Contribution:** 3
**Rating:** 8
**Confidence:** 4

**Summary:**

The paper illustrates a synthetic data generation pipeline, SynPO, for self-generating preference data to align the model. The pipeline consists of (a) self-generating diverse prompts at scale and (b) self-generating paired responses for these prompts. The paired responses are constructed by (i) first sampling generations from the base model, (ii) training a response improver model to synthetically generate the preferred response, and (iii) pairing the preferred response with the base model response to create paired preference data. The method relies only on initial high-quality seed SFT data, and all the remaining data is synthetic. Applying the SynPO method for multiple iterations results in significant alignment performance gains.

**Strengths:**

- Each component of the SynPO pipeline is intuitive and contributes to the final model performance
- Strong and extensive empirical results, including data analysis (Figure 4, 5) and extensive ablation experiments.
- Creative uses of the seed SFT data for multiple components of the pipeline.
- Very well written and structured.

**Weaknesses:**

- Has the synthetic preference data been tested for benchmark data leakage? I didn't see anything suggesting that checks for data leakage were done.
- While there's novelty to the pipeline as a whole, the individual components of the pipeline have been known.

There are a few typos as well:
- L193: "an infinite" -> "a huge" (clearly what is being described is not "infinite")
- L247: \sigma -> \beta. Also, state what is \sigma? I assume it's the sigmoid function.
- L314: "involve" -> "compare against"
- L316: "involves" -> "use"

**Questions:**

Data filtering:
- For preference data, why is the rejected response across iterations the one generated by the base model? Why are the model(s) from the previous iteration(s) not used for generating rejected responses?
- Given that the "chosen response" quality keeps improving in the preference data since the rejected response is from the base model, do the number of filtered-out preference pair data, as stated on L230, come down across iterations?

Results:
- I don't understand the performance gains on TruthfulQA. Any reason for/pattern to these gains?

---

> ### Author Response · Authors · 2024-11-21
> **Reply to Reviewer EKkD (Part 1/2 )**
>
> We sincerely thank Reviewer EKkD for the positive recommendation as well as the valuable suggestions. We really appreciate your kind words that our work is intuitive and has good results. Below we would like to give detailed responses to each of your comments.
>
> **Q1: “Has the synthetic preference data been tested for benchmark data leakage? I didn't see anything suggesting that checks for data leakage were done.”**
>
> Thank you for this insightful point. As suggested, we conducted experiments to test for benchmark data leakage.
>
> Specifically, we compared the n-grams of our training datasets (seed SFT data, synthetic preference data, and UltraFeedback for reference) with the n-grams of the test set data to identify any overlaps. If any n-gram from a test data entry appears in our dataset n-grams, that entry is marked as leaked, and we calculate the proportion of leaked data for each test dataset. For datasets with candidate answers, we concatenate the question with candidate answers for analysis; for those without candidate answers, we use only the question. Following HELM\[1\], we set the n-gram size to 13.
>
> The results are as follows:
>
> | Data | Arc  | HellaSwag  | TQA  | MMLU  | Winogrande | GSM8k  |  |
> | :---- | :---- | :---- | :---- | :---- | :---- | :---- | :---- |
> | UltraFeedback (for reference) | 0.00085 | 0.00030 | 0.00122 | 0.00199 | 0.00000 |  0.00531 |  |
> | Seed SFT Data | 0.00085 | 0.00010 | 0.00122 | 0.00036 | 0.00079 | 0.00076 |  |
> | Synthetic Preference Data | 0.00000 | 0.00000 | 0.00122 | 0.00064 | 0.00000 | 0.00000 |  |
> |
>
> *(Benchmark data leakage test for Open LLM Leaderboard tasks. UltraFeedback data for reference.)*
>
> | Data | AlpacaEval 2.0  | Areana-Hard  | MT-Bench |
> | :---- | :---- | :---- | :---- |
> | UltraFeedback | 0.00248 | 0.00600 | 0.01250 |
> | Seed SFT Data | 0.00373 | 0.01200 | 0.01250 |
> | Synthetic Preference Data | 0.00124 | 0.00800 | 0.00000 |
> |
>
> *(Benchmark data leakage test for instruction-following benchmarks. UltraFeedback data for reference.)*
>
>
>
> Overall, the overlap between our training datasets (seed data and synthetic preference data) and the test sets is very low, indicating that there is no data leakage issue.
>
> Interestingly, **synthetic preference data generally shows even less overlap with the test sets than other data**.
>
> We will incorporate these findings into the revised version of our paper. Thank you for highlighting this important aspect, which certainly enhances the robustness of our research.
>
> **Q2: “While there's novelty to the pipeline as a whole, the individual components of the pipeline have been known.”**
>
> The continuous improvement of LLMs has  always been a challenging and complex process \[2,3,4\]. In our setting, it mainly involves self-prompt generation, response synthesis, and optimization.
> - Self-prompt generation: our self-prompt generation method differentiates itself from previous approaches that rely on seed data or larger models by enabling the model to **self-generate** high-quality, diverse prompts based on **random keywords from pretraining corpora**.
> - Response synthesis: to the best of our knowledge, we are the first to **utilize pre- and post-self-refinement responses to construct synthetic preference pairs**.
>
> While we adopt SimPO’s loss function during the optimization phase, the innovations in the first two parts are significant contributions to prompt generation and synthetic preference data.
>
> **Q3: Typos**
>
> Thank you for your meticulous review and for pointing out the typos. Yes, on L247, the (\\sigma) should indeed be (\\beta). We will update the mentioned typos in the latest version.

---

> > ### Author Response · Authors · 2024-11-21
> > **Reply to Reviewer EKkD (Part 2/2 )**
> >
> > **Q4: “Why is the rejected response across iterations the one generated by the base model? Why are the model(s) from the previous iteration(s) not used for generating rejected responses?”**
> >
> > Yes, using the outputs of the model from the previous iteration as rejected responses is a natural approach and was indeed our initial attempt. However, we found that this approach did not yield good results. Specifically, starting from the second round, each iteration would drop by 1.2 to 2.6 points on the AplacaEval 2.0 benchmark compared to using the one generated by the base model (using Llama 3B). This issue arises from the preference optimization loss derived from SimPO\[5\] (in L241-247 of our paper).
> >
> > When we mix the preference data from the (t-1) and (t) iterations, the outputs from the (t-1) model appear as $y^l$ in half of the data, requiring a reduction in probability, but as $y^w$ in the other half, requiring an increase in probability. This causes a conflict in learning, and consistently regarding the initial model outputs as rejected ones avoids this contradiction.
> >
> > **Q5:  Do the number of filtered-out preference pair data, as stated on L230, come down across iterations?**
> > Yes, as the quality of model generation improves with each SynPO iteration, the number of filtered-out preference pairs decreases. In our experiments, we randomly integrated 10,000 preference pairs from each iteration into the overall synthetic preference dataset (L911).
> >
> > **Q6: Explanation for the performance gains on TruthfulQA**
> >
> > There are two main reasons for the observed performance gains on the TruthfulQA benchmark:
> >
> > 1. **Correlation between preference alignment and TruthfulQA::** TruthfulQA assesses the truthfulness of responses from language models, a goal that aligns closely with preference alignment. Our synthetic preference dataset, which includes instances emphasizing truthfulness, enhances the model's ability to accurately interpret context and generate truthful responses. This aligns with findings from the Meng et al.\[5\], which states:
> > >  "Preference optimization methods consistently improve TruthfulQA performance, with some enhancements exceeding 10%. Similarly, we hypothesize that the preference dataset contains instances that emphasize truthfulness, which helps the model better understand the context and generate more truthful responses."
> >
> > 2. **Utilization of SimPO Loss:** We employ SimPO loss for preference optimization, which has significantly boosted TruthfulQA performance compared to other methods, as detailed in Table 9 of the SimPO paper\[5\].
> >
> >
> > It's also worth noting that aside from TruthfulQA, the SynPO approach has led to significant improvements in other tasks. For instance, on EQ-Bench, we observed an improvement from a score of 46.79 to 55.82 with the Llama3-SynPO-4iter model. This alignment is particularly beneficial for tasks that involve safety and helpfulness, demonstrating the broad applicability of our approach.
> >
> > Overall, we greatly appreciate your efforts for your thoughtful and insightful comments on our paper. We would be happy to do any follow-up discussion or address any further comments.
> >
> > **Reference:**
> > *\[1\] Liang P, Bommasani R, Lee T, et al. Holistic evaluation of language models\[J\]. arXiv preprint arXiv:2211.09110, 2022\.*
> >
> > *\[2\] Yuan, Weizhe, et al. "Self-rewarding language models." arXiv preprint arXiv:2401.10020 (2024)\.*
> >
> > *\[3\]  Wu, Tianhao, et al. "Meta-rewarding language models: Self-improving alignment with llm-as-a-meta-judge." arXiv preprint arXiv:2407.19594 (2024).*
> >
> > *\[4\] Yin, Yueqin, et al. "Self-Augmented Preference Optimization: Off-Policy Paradigms for Language Model Alignment." arXiv preprint arXiv:2405.20830 (2024).*
> >
> > *\[5\] Meng, Yu, Mengzhou Xia, and Danqi Chen. "Simpo: Simple preference optimization with a reference-free reward." arXiv preprint arXiv:2405.14734 (2024).*

---

> > > ### Comment · Reviewer_EKkD · 2024-11-24
> > >
> > > I appreciate the detailed responses to my queries.
> > >
> > > It's great to see that the gains can't be attributed to data leakage. However, my only qualm with the analysis is that it should be embedding-based rather than n-gram-based - [lmsys blog](https://lmsys.org/blog/2023-11-14-llm-decontaminator/)
> > >
> > > While not the main focus of this work, it will be great to demystify the reasons for the gains on TruthfulQA with SynPO/SimPO.

---

> > > > ### Author Response · Authors · 2024-11-26
> > > > **Response to Reviewer EKkD**
> > > >
> > > > Many thanks for your positive feedback on our paper and your response to our rebuttal.
> > > >
> > > > To further test for data leakage, we conducted the embedding-based check as suggested \[1\] (and we are very grateful for pointing us towards this elegant approach). We utilized the GPT-4-Turbo API to report the contamination percentage (%) on the test set, with UltraFeedback results serving as a reference. For a quick check, we randomly selected 1000 samples from test sets larger than 1000 samples (using the full set for those with fewer than 1000 samples) for experimentation.
> > > >
> > > > The results are as follows:
> > > >
> > > > | Data | Arc  | HellaSwag  | TQA  | MMLU  | Winogrande | GSM8k  |
> > > > | :---- | :---- | :---- | :---- | :---- | :---- | :---- |
> > > > | UltraFeedback (for reference) | 2.2% | 1.5% | 2.1% | 0.4% | 1.2% | 0.0% |
> > > > | Seed SFT Data | 1.4% | 1.6% |0.5% | 0.3% | 1.1% | 0.0% |
> > > > | Synthetic Preference Data | 1.0% | 0.6% | 0.9% | 0.3% | 0.2% | 0.0% |
> > > > |
> > > >
> > > > *(Embedding-based data leakage test results on LLM Leaderboard tasks.)*
> > > >
> > > >
> > > > | Data | AlpacaEval 2.0  | Arena-Hard  | MT-Bench |
> > > > | :---- | :---- | :---- | :---- |
> > > > | UltraFeedback | 5.3% | 1.8% | 1.5% |
> > > > | Seed SFT Data | 4.5% | 2.0% | 1.6% |
> > > > | Synthetic Preference Data | 3.9% | 1.6% | 1.2% |
> > > > |
> > > >
> > > > *(Embedding-based data leakage test results on three instruction-following benchmarks.)*
> > > >
> > > > Overall, the conclusion remains that the gains cannot be attributed to data leakage. In comparison, embedding-based detection can identify more instances of semantic overlap, making it a more effective method of detection. Also, synthetic preference data generally shows even less overlap with the test sets than other data.
> > > >
> > > >
> > > > We agree that demystifying the reasons for the gains on TruthfulQA will improve the clarity. As suggested, we will incorporate explanations for the performance gains on TruthfulQA in our final version.
> > > >
> > > > Again, thank you again for your detailed reviews and valuable insights\!
> > > >
> > > >
> > > > **Reference:**
> > > > *\[1\] Yang S, Chiang W L, Zheng L, et al. Rethinking benchmark and contamination for language models with rephrased samples\[J\]. arXiv preprint arXiv:2311.04850, 2023\.*

---

> > > > > ### Comment · Reviewer_EKkD · 2024-11-27
> > > > >
> > > > > Wonderful! I think this additional analysis would make the paper quite strong.
> > > > > I really appreciate the quick and precise turnaround by the authors.

---

### Official Review · Reviewer_TAJC · 2024-11-08

**Soundness:** 2
**Presentation:** 2
**Contribution:** 2
**Rating:** 5
**Confidence:** 3

**Summary:**

The paper presents "SynPO", which trains a self-prompt generator by controlling the keywords used. Noises (extracted also from responses) are inserted to produce the "bad" responses so that a SFT setup is possible where good and bad responses are now available. A separate step involves a response regenerator that is used to refine the model responses to get the good responses.
The diversity comes from the selection of keywords, which can be sampled from a large pretraining set like RefinedWeb, and the model improves by learning to discern bad and good responses iteratively.

**Strengths:**

The strength of the paper is that it offers a sound solution in terms of introducing diversity in the self-improvement process. While this is not exactly novel, but I do see a world where this works and improves model progressively. Moreover, use of keyword means that it provides a transparency in what we can control, the granularity of control, and ease of analysis.

**Weaknesses:**

The weaknesses are the following:
1) I'm not sure if this is completely "self-boosting" as the title claimed, since external knowledge is needed, and careful selection of the keyword might be needed to guide the LLM to learn properly.
2) I think the paper does not provide much insights into the types of noise (keywords) that are more effective, which seems rather important as an insights in creating the bad samples.
3) The method relies heavily on the model's ability to generate out from a prompt containing keyword, it might or might not work with less capable LLMs.
4) The writing is rather unclear, I think it could have been written in a much more understandable manner.

**Questions:**

Do you have any insights on what kind of noise keywords are bad? What exactly does it make one keyword more noisy than another?

---

> ### Author Response · Authors · 2024-11-20
> **Reply to Reviewer  TAJC (Part 1/2 )**
>
> We sincerely thank Reviewer TAJC for the review and are grateful for the time you spent with our submission. We are glad for the acknowledgement that our approach is sound and provides transparency for control and analysis. We wish to address your concerns by giving detailed responses to each of your comments as follows:
>
> **Q1: The self-boosting mechanism**
>
> Yes, a small amount of seed data is required.
>
> - Compared to "self-improvement," we use the term "self-boosting" to emphasize that the model's continuous iterative enhancement is **self-driven (by the self-refinement process and pre- and post-refinement contrast)** rather than claiming to be entirely independent. We aim to highlight that, unlike previous works that rely on external tools or teacher LLMs[1][2], SynPO is mainly a process of self-correction and self-driven improvement.
> - Additionally, we would like to clarify that the **external knowledge required by our method is minimal**, similar to the self-rewarding approach[3], including a small amount of seed data.
> - Besides, the selection of keywords is **entirely automatic (simple rule-based) and does not require careful selection or human effort**.
>
>
> **Q2: Concerns and questions about bad noising keywords**
>
> There seems to be some misunderstandings on the motivation of noise keywords introduction and our insights in creating the rejected samples.
>
> - First, we would like to clarify that the introduction of noise keywords is merely an **optional trick to enhance the robustness** of the prompt generator (L155-158). For each prompt $x^*_i$ in the seed data, we randomly extract two keywords from $x^*_i$ and one noise keyword from $x^*_j, j ≠ i$. This ensures that the training data output excludes semantically irrelevant input keywords, guiding the model to generate prompts based on relevant keywords while disregarding unrelated words in the given list. Therefore, this strategy **aims to ensure the naturalness and semantic coherence of the generated prompt, not to produce bad samples**.
> - Since the noise keywords serve only to introduce noise, there is no need for an additional strategy to ensure one keyword is more noisy than another. Simply sampling randomly from the entire keyword list, excluding the current prompt's keywords, is sufficient.
>
> To further demonstrate that noise keywords are an optional strategy in our method, we have conducted experiments to show their impact. We trained and evaluated the Llama3 prompt generator with three random keywords, either including or excluding one noise keyword. We calculated the average similarity across the generated prompts using Sentence-Transformer, as mentioned in Section 2.2, and employed GPT-4 Turbo for LLM-as-a-Judge evaluation of prompt quality (on a scale of 1 to 10, where 1 represents a prompt that is very unrealistic, unnatural, or unanswerable, and 10 represents a prompt that is very reasonable, realistic, and answerable).
>
>
> | Noise Condition | Avg. Similarity | Quality |
> |-----------------|----------------|---------|
> | no noise        | 0.0572         | 7.92   |
> | w noise         | 0.0574          | 8.99   |
> |
>
> *(Evaluation run on 1k data for each setting. Each example contains 3 keywords (3 keywords for the 'no noise' setting, 2 keywords and 1 noise keyword for the 'w noise' setting)*
>
> The results are listed in the following table. They show that the inclusion of noise keywords improves the quality of the self-generated prompts (as LLMs learn to ignore unrelated words), but **even without noise keywords, the self-prompt generator can still generate relatively high-quality prompts**.

---

> ### Author Response · Authors · 2024-11-20
> **Reply to Reviewer TAJC (Part 2/2 )**
>
> **Q3: Which type of keywords are more effective**
>
> As clarified, the noise keywords maintain the same type as the extracted keywords for better robustness. In our implementation, the keywords are randomly selected from rule-based filtered phrases. Specifically, we use the NLTK toolkit to filter out stop words, extract all noun phrases from the sentences, and remove any preceding articles. We then randomly sample from these phrases. The decision to use noun phrases is based on the following preliminary experiments:
>
> |  Extraction Form | avg similarity | Quality |  |
> | :---- | :---- | :---- | :---- |
> | Noun Phrases | 0.0574 | 8.99 |  |
> | Verb Phrases | 0.0865 | 8.74 |  |
> | Noun \+ Verb Phrases | 0.0610 | 8.67 |  |
> | All Phrases | 0.0569 | 8.67 |  |
> | Noun Words | 0.0604 | 8.98 |  |
> | Verb Words | 0.0894 | 8.76 |  |
> | Noun \+ Verb Words | 0.0725 | 8.52 |  |
> | All Words | 0.0598 | 8.61 |   |
> |
>
> *(Note: "Words" refer to single-word keywords, while "Phrases" refer to each keyword and can be a phrase comprising 1 to 3 words.)*
>
> As shown, using phrases rather than limiting keywords to single words generally results in better diversity and quality. This improvement may be due to phrases incorporating more inductive bias from the pretraining data for prompt generation. Specifically, while random sampling from all phrases or words without filtering is acceptable, it is not as effective as random sampling from nouns, as nouns contain the most important information needed to diversify a sentence.
>
> Beyond these evaluation results, we have also manually observed the impact of different extraction forms and found that using phrases as keywords generally resulted in higher quality prompts. Therefore, our experiments utilize randomly sampled noun phrases. We will include these preliminary experiments, which support our choice of keyword extraction methods, in the Appendix to provide further insights into self-prompt generator training.
>
>
> **Q4: New insights in creating the bad samples**
> As clarified in Q2, we do not regard prompts with noise keywords and their corresponding responses as bad samples. The synthetic prompts are the same for both chosen and rejected samples. Our main intuition in constructing good and bad samples lies in **inducing pre- and post-self-improvement responses as natural rejected and chosen candidates, respectively** (Section 2.2). The chosen responses provide clear guidance on what approximates a gold standard response through the iterative self-refinement process. This method of generating synthetic preference is fundamentally different from previous methods that sample multiple responses from a model and then score them as good or bad samples \[1\]. We sincerely hope that this clarification addresses your concerns regarding the novelty of our insights.
>
>
> **Q5: Prompt generation capability of less capable LLMs**
> The prompt generation capability of an LLM can be influenced by the model's scale. However, since our method synthesizes training data through the construction of keywords to prompts, even a 0.5B model can be finetuned to be a good self-prompt generator. To further illustrate this point, we conducted experiments using the Qwen2.5 model with 0.5B and 1.5B parameters:
>
> | Model | Avg. Similarity | Quality |  |
> | :---- | :---- | :---- | :---- |
> | Llama3-8B | 0.0574 | 8.99 |  |
> | Qwen2.5-Instruct-1.5B | 0.0602 (+0.0028) | 8.25 (-0.74) |  |
> | Qwen2.5-Instruct-0.5B | 0.0617 (+0.0043) | 8.03 (-0.96) |  |
> |
>
> *( Evaluation run on 1k data for each setting )*
>
> It can be observed that, consistent with characteristics of LLMs, the prompt generation capability is also affected by the model size. However, even **a small model with 0.5B parameters can become a good self-prompt generator after training with our method** (generating prompts with high quality and diversity).
>
> On the other hand, we focus on the continuous improvement of LLMs. Thus, self-boosting primarily addresses the shortage of high-quality preference data. Our experiments mainly use 7B and 8B models, which are moderately sized. **For very weak small models, further enhancement can be easily achieved by using a stronger teacher model to construct distillation data.** Even a slightly larger teacher model can provide adequate supervision at a low cost.
>
> Overall, many thanks for your insightful points and suggestions. These comments really help improve our work. We hope our answers have addressed your concerns. If you have any further questions, we are happy to address them.
>
> **Reference:**
> *\[1\] Li, Haoran, et al. "Synthetic data (almost) from scratch: Generalized instruction tuning for language models." arXiv preprint arXiv:2402.13064 (2024).*
> *\[2\] Shi, Taiwei, Kai Chen, and Jieyu Zhao. "Safer-instruct: Aligning language models with automated preference data." arXiv preprint arXiv:2311.08685 (2023).*
> *\[3\] Yuan, Weizhe, et al. "Self-rewarding language models." arXiv preprint arXiv:2401.10020 (2024).*

---

> > ### Author Response · Authors · 2024-11-24
> > **Further comments and discussions will be appreciated!**
> >
> > Dear Reviewer TAJC,
> >
> > Thank you for your valuable time to review our work and for your constructive feedback. We posted our response to your comments four days ago, and we wonder if you could kindly share some of your thoughts so we can keep the discussion rolling to address your concern if there are any.
> >
> > In the previous response,
> >
> > 1. As suggested, we added our preliminary experiments on keywords analysis in the Appendix H which support our choice of keyword extraction methods, to provide further insights into self-prompt generator training.
> >
> > 2. We clarified that noise keywords are an optional strategy for the prompt generator. Our method does not generate bad samples but rather induces pre- and post-self-improvement responses as natural rejected and chosen candidates. This approach is fundamentally different from previous methods.
> >
> > 3. We explained that while a small amount of seed data is required, the term "self-boosting" emphasizes the self-driven nature of the model's iterative enhancement. The external knowledge required is minimal, and the selection of keywords is automatic and rule-based.
> >
> > 4. We demonstrated that even less capable LLMs can be fine-tuned to be good self-prompt generators. It is worth mentioning that even a 0.5B model can generate high-quality prompts after training with our method.
> >
> > We would appreciate it if you could kindly take a look at both the revision and our response to your comments. If you have any further questions, we are happy to discuss them!
> >
> > Best regards,
> >
> > Authors

---

> > > ### Author Response · Authors · 2024-11-25
> > > **Follow up to reveiwer TAJC**
> > >
> > > Dear Reveiwer TAJC,
> > >
> > > We would like to thank you again for your detailed reviews. We have updated our draft and added replies to your Cons with our latest experimental results.
> > >
> > > Since the rebuttal deadline is approaching soon, a lot of papers have finished the discussion. Given that your current score is 5, we would appreciate it if you could let us know if our responses have addressed your concerns satisfactorily. If your concerns have not been resolved, could you please let us know about it so that we have the opportunity to respond before the deadline?
> > >
> > > We would be happy to have any follow-up discussions or address any additional concerns.
> > >
> > > Thanks very much! Looking forward to your reply.
> > >
> > > Best,
> > >
> > > Authors

---

> > > > ### Author Response · Authors · 2024-11-27
> > > >
> > > > Dear Reviewer TAJC,
> > > >
> > > > Thank you once again for your thorough reviews. We have revised our draft and included responses to your questions.
> > > >
> > > > As the rebuttal deadline is nearing, we would greatly appreciate it if you could confirm whether our responses have adequately addressed your concerns. If they have, we would be thankful if you could consider increasing your score. We are open to further discussions or addressing any additional points you may have. Thank you very much!
> > > >
> > > > Best,
> > > >
> > > > Authors

---

### Author Response · Authors · 2024-11-22
**General Response to the Reviewers**

We sincerely thank all the reviewers for their great efforts in reviewing our paper and for their constructive comments, which clearly helped us strengthen our paper.  We are encouraged to find that the reviewers appreciate the novelty and intuitiveness of SynPO (Reviewer EKkD, Ftfv), the sound solution introducing diversity in the self-improvement process (Reviewer TAJC), the extensive and solid experimentation (Reviewer EKkD, LmCF), and the clear presentation quality (Reviewer EKkD).

We have followed the helpful suggestions from all reviewers, and updated the additional experiments and discussion in the new version. Here, we make a brief summary of the changes made in the updated version:


1. We have included preliminary experiments on keyword analysis in Appendix H (Reviewers TAJC, d9yw), which support our choice of keyword extraction methods and provide deeper insights into the training of the self-prompt generator.
2. We have revised paragraph 3 in the introduction for improved clarity (Reviewer LmCF) and corrected typos in the main text (Reviewer EKkD).
3. We have expanded the discussion on the mentioned works in Section 5 (Reviewer d9yw) in the revised version.
4. We have conducted experiments to test for benchmark data leakage in Appendix I (Reviewer EKkD). It is worth mentioning  that synthetic preference data shows even less overlap with test sets compared to other data.
5. In our original paper, we evaluated on three instruction-following benchmarks (AlpacaEval 2.0, ArenaHard, MT-Bench), six diverse LLM Harness tasks, and the well-recognized Open LLM Leaderboard. In the revised version, we have expanded our evaluation to include ten additional tasks (including reasoning tasks and domain-specific tasks) and report domain-specific scores on AGIEval in Appendix J  (Reviewer Ftfv, d9yw).

Once again, we thank all reviewers for their valuable feedback. We are happy to continue the discussion if there are any further questions.

---

### Meta-Review · Area_Chair_tZze · 2024-12-19

**Metareview:**

This paper introduces SynPO, a self-boosting framework that leverages synthetic preference data to iteratively improve the performance of large language models (LLMs). The approach trains a self-prompt generator and response improver to produce synthetic prompts and refine responses, eliminating the need for large-scale human annotation. The method is examined  on Llama3-8B and Mistral-7B, showing significant improvements in alignment tasks (e.g., AlpacaEval 2.0) and general benchmarks (e.g., MMLU, TruthfulQA).

Strengths: The paper addresses a critical challenge of lack of high-quality preference data by introducing a scalable and iterative synthetic data generation approach. The experiments are extensive, with strong empirical results supported by ablation studies and analyses of key components (e.g., noise keywords, extraction forms). The method demonstrates strong applicability to alignment tasks with clear presentations.

Weaknesses: The paper lacks detailed discussion on out-of-distribution robustness, and while the authors argue for the effectiveness of synthetic data, additional analyses on unexplained performance gains (e.g., on knowledge benchmarks) could improve the work. The introduction and framing of key components (e.g., model roles) were initially unclear, though this was improved during the rebuttal period.

I recommend accepting this paper for its scalable method, solid empirical results and clear presentation.

**Additional Comments On Reviewer Discussion:**

During the rebuttal period, several key points were raised and addressed by the authors:

1. The proposed self-boosting method relies on external data:

The authors clarified that "self-boosting" refers to the iterative self-improvement mechanism, with minimal external input required. And they use automatic keyword selection.

2. Concerns were raised about the improvement on knowledge benchmarks and potential in-distribution biases. The authors expanded evaluations to broader domains (e.g., AGIEval) and provided embedding-based data leakage checks, which verifies minimal overlap and robustness.

3. Reviewers want to see more ablation studies to understand why synthetic data outperformed alternatives. The authors provided experiments comparing synthetic and real data, showing the advantages of synthetic data in diversity and iterative learning.

4. Confusion in the introduction about model roles was noted. The authors revised this section for clarity.

Overall, authors addressed reviewers' concerns well.

---

### Decision · Program_Chairs · 2025-01-22

Accept (Poster)